# Development of a 3D tracking system for multiple marmosets under free-moving conditions
Terumi Yurimoto [1], Wakako Kumita[1], Kenya Sato[1], Rika Kikuchi[1], Gohei Oka[1], Yusuke Shibuki[1], Rino Hashimoto[1], Michiko Kamioka[1], Yumi Hayasegawa[1], Eiko Yamazaki[1], Yoko Kurotaki [2], Norio Goda[3], Junichi Kitakami[4], Tatsuya Fujita[5], Takashi Inoue [1] & Erika Sasaki [1] ✉

Assessment of social interactions and behavioral changes in nonhuman primates is useful for understanding brain function changes during life events and pathogenesis of neurological diseases. The common marmoset (*Callithrix jacchus*), which lives in a nuclear family like humans, is a useful model, but longitudinal automated behavioral observation of multiple animals has not been achieved. Here, we developed a Full Monitoring and Animal Identification (FulMAI) system for longitudinal detection of three-dimensional (3D) trajectories of each individual in multiple marmosets under free-moving conditions by combining video tracking, Light Detection and Ranging, and deep learning. Using this system, identification of each animal was more than 97% accurate. Location preferences and inter-individual distance could be calculated, and deep learning could detect grooming behavior. The FulMAI system allows us to analyze the natural behavior of individuals in a family over their lifetime and understand how behavior changes due to life events together with other data.

Behavioral analyses are important in broad research areas such as animal and human biological, psychological, and ethological studies. In neuroscience research, behavioral analysis is one of the most important analytical methods for understanding changes in brain function during development and aging and assessing disease development in animal models. In particular, nonhuman primates are often used to study brain function, and behavioral characteristics provide an indispensable source of data for hypothesis testing[1]. The common marmoset (*Callithrix jacchus*) is a small nonhuman primate with behavioral and social characteristics resembling those of humans. They are diurnal, form monogamous families, engage in altruistic behaviors such as food sharing, and cooperate with all family members to raise their youngest offspring[2–4]. As marmoset body size and social units are small, maintaining similar to social units in the wild for research laboratory is easier than that for other primates. Therefore, marmosets are an ideal model for studying social behaviors in social units[2].

To evaluate long-term behavioral changes due to physiological changes, such as development, aging, and progression of disease using marmosets, it is important to capture changes in behavior over a lifespan, and an analysis of multiple marmosets under free-moving conditions is necessary to capture changes in social behavior. Therefore, a novel system is required to detect and quantify the individual behaviors of multiple marmosets under free-moving conditions in real time, both automatically and accurately. Automated behavior analysis in a home cage enables observations of more natural animal behavior[5–7]. This analysis has the advantages of being able to ignore environmental condition, monitor social behavior in multiple animals, and analyze changes in social behavior over a relatively long period of time[8].

The various systems used for analyzing animal behaviors under free-moving activities include systems suitable for three dimensions (3D) tracking and behavior classification (DANNCE[9], FreiPose[10], and MarmoDetecotor[11]), for generating pose estimation (DeepLabCut, SLEAP[12], MARS[13], DANNCE, FreiPose, and OpenMonkeyStudio[1]), for analyzing behavior in home cages (DeepLabCut[14,15], B-SoiD[8], SLEAP, and MARS), for analyzing multiple animals (DeepLabCut, MiceProfiler[16], SLEAP, and OpenMonkeyStudio), for finding automatic classification of behavior by unsupervised clustering (MoSeq[17], B-SoiD, and FreiPose[10]), and for use across animal species (DeepLabCut, DeepBhvTracking[18], B-SoiD, and DANNCE). These systems have been generally developed for mice; only DeepLabCut, DeepBhvTracking, DANNCE, and MarmoDetector[11] have

[1]Department of Marmoset Biology and Medicine, Central Institute for Experimental Medicine and Life Science, Kawasaki 210-0821, Japan. [2]Center of Basic Technology in Marmoset, Central Institute for Experimental Medicine and Life Science, Kawasaki 210-0821, Japan. [3]Public Digital Transformation Department, Hitachi, Ltd., Shinagawa 140-8512, Japan. [4]Vision AI Solution Design Department Hitachi Solutions Technology, Ltd, Tachikawa 190-0014, Japan. [5]Engineering Department Eastern Japan division, Totec Amenity Limited, Shinjuku 163-0417, Japan. ✉e-mail: esasaki@ciea.or.jp

been adapted to marmosets, which move relatively quickly in 3D. A previous study used the pose estimation system to analyze the behavior of two marmosets for 9 h using individual identification markers[14]. These analysis systems have observed only a small proportion of an animal's life and longitudinal observation over each marmoset's lifetime (15 years) or more than months of observing a family have not been achieved. This is possibly because automatically detecting marmoset trajectories, behaviors, and interactions between individuals under free-moving conditions in multiple animals is especially challenging. For example, marmosets are often close to each other when grooming, mating, or fighting[19]. This proximity makes it difficult to detect individual bodies, which complicates individual tracking and behavior detection.

In this study, we developed a behavioral analysis system named FulMAI (Full Monitoring and Animal Identification), that can record and analyze behavior of an animal living among multiple animals over their lifetime. The concept of this system is to detect changes in behavior over an individual's lifespan and understand the relation of changes in brain function. FulMAI is a 3D tracking system that uses cameras, light detection and ranging (Lidar) devices, and deep learning to simultaneously track a marmoset family of three individuals in one cage. Furthermore, to take advantage of stress-free behavior analysis in the home cage, we also developed a system for individual identification via facial recognition using deep learning. The FulMAI system can analyze the location and the period each animal stayed in the cage and the distance between individuals. Furthermore, the time spent in social behavior is an index for analyzing animal interactions. Therefore, as an example of social behavior detection, we developed automatic grooming behavior detection system using deep learning. The technology developed in this study would be applicable in detecting important behavioral changes caused by development, aging, and various diseases in small, nonhuman model primates.

## Results

### Marmoset facial recognition by deep learning

Five marmosets of the same generation (1–2 years old) were used to examine whether deep learning for face identification could be adapted to marmosets. These marmosets were kept in different cages in a conventional breeding colony and facial images were collected individually. To evaluate the trained models, other 500 facial images (100 for each animal) were obtained from the videos not used for training. The evaluation metrics for deep learning were as follows: specificity of 99.8–100%, accuracy of 99.4–99.8%, recall of 98.0–100%, precision of 99–100%, and an F-measure of 98.5–99.5% (Table 1). The overall identification accuracy (total number of correct images/total number of images) of the five marmosets was 99.2%.

Subsequently, face identification accuracy was examined in a family of three, that was raised in the set-up home cage. The trained model was evaluated using data that were not used for training. For the evaluation, 5600 facial images (1400 for each class), three classes including each marmoset, and an unknown class were used. The unknown class was used to exclude inappropriate images, such as unclear images, images of the back of the head and faces in profile, and images not showing faces. The evaluation metrics for deep learning for individual identification were as follows: specificity of 98.7–99.7%, accuracy of 98.6–99.2%, recall of 96.1–98.8%, precision of

98.4–99.1%, and an F-measure of 97.3–98.3% (Table 2). The overall percentage of correct identifications was 98.0%.

### Multiple animals' 3D tracking

To optimize the Lidar for laser measurement and video recording, a part of the cage made of metal mesh was replaced with acrylic panels, following which suitable measurement results were obtained (Fig. 1a). The laser beam distance needed to be sufficiently narrow and cover the entire cage; thus, the Lidars were placed 1000 mm from the front of the cage (Fig. 1b). The overview of this system workflow is shown in Fig. 1c. The videos and Lidar data were processed on the same PC, and acquisition times were perfectly synchronized for at least 1 month. Yolo, an object recognition algorithm, detected the marmoset's face, body, actions, and environmental enrichments (ball and hammock) in each video frame. Body coordinates were combined with coordinates using Yolo and Lidar to calculate a short 3D trajectory. Furthermore, the cropped face images were identified as individuals using a convolutional neural network, and individual IDs were applied to the short-term 3D trajectories to achieve long-term (average 2.8 min.) 3D tracking of each individual. Behaviors were detected simultaneously with the same Yolo model, and each animal's behavior was linked to the nearest marmoset ID. All 3D tracking and grooming behavior detection tests were conducted using data from one family consisting of three marmosets (I5072M/father (marmoset A), I5894F/mother (marmoset B), and I940M/ juvenile (marmoset C)) in a cage.

The 3D coordinate information of marmosets was obtained using video tracking and Lidar (Fig. 2a–f, and Supplementary Movie 1). While the spatial resolution of Lidar was not enough to detect each marmoset and its behavior, it was sufficient to detect the body and tail. Therefore, the centroids of the point clouds were used as animal locations for 3D trajectory.

A raw image taken by the upper left video camera captured the entire cage and the three marmosets (Fig. 2a), and the data was acquired by Lidar simultaneously (Fig. 2b). The centroid of cluster by Lidar was indicated in white points (Fig. 2b), and each point was given a sequential tracking ID number during background processing. A new number was assigned each time when the trajectory was broken. The 2D pixel locations of the marmosets were then detected using an object detection algorithm (Yolov3[20]) in the video image (Fig. 2c). The same process was applied to the four cameras to determine the locations of the marmosets for each camera. The 3D location of each marmoset was calculated using pre-calibrated camera positions and the pixel coordinates of the marmosets (Fig. 2d). The extracted face images of marmosets by Yolo were identified using the trained VGG19 model, and individual IDs were assigned to each marmoset by facial recognition (Fig. 2e, f). These series of processed video data are shown in Supplementary Movie 1. In some cases, the marmosets could not be recognized by video tracking when they moved too fast or behind an object. In such cases, the trajectory was interpolated by Lidar (Supplementary Movie 2, 3). Figure 3 shows 5 min of tracking trajectories without face identifications (Fig. 3a, b) and with face recognition (Fig. 3c). When Lidar alone or Lidar and video tracking were performed, the tracking serial numbers changed frequently, and it was impossible to continuously track the individual (Fig. 3a, b). However, by combining face recognition with Lidar and video tracking, the trajectory of each animal was continuously detected, and marmoset IDs were assigned (Fig. 3c).

**Table 1 | Classification accuracy of face identification in five marmosets aged 1 to 2 years**

| | Animal ID | | | | |
| --- | --- | --- | --- | --- | --- |
| | I774 | I6695 | I6708 | I6677 | I893 |
| Specificity | 100.0% | 99.8% | 99.8% | 99.8% | 99.8% |
| Accuracy | 99.8% | 99.6% | 99.4% | 99.8% | 99.8% |
| Recall | 99.0% | 99.0% | 98.0% | 100.0% | 100.0% |
| Precision | 100.0% | 99.0% | 99.0% | 99.0% | 99.0% |
| F-Measure | 99.5% | 99.0% | 98.5% | 99.5% | 99.5% |

**Table 2 | Classification accuracy of face identification in a marmoset family**

| | Marmoset A | Marmoset B | Marmoset C | Unknown |
| --- | --- | --- | --- | --- |
| Specificity | 98.7% | 99.5% | 99.7% | 99.5% |
| Accuracy | 98.7% | 98.6% | 99.2% | 99.5% |
| Recall | 98.8% | 96.1% | 97.6% | 99.6% |
| Precision | 96.1% | 98.4% | 99.1% | 98.5% |
| F-Measure | 97.4% | 97.3% | 98.3% | 99.0% |

**Fig. 1 | Installation of hardware and software for the FulMAI system. a** A home cage with acrylic panels on the front and back and metal mesh on the sides. **b** Schematic diagram of light and ranging (Lidar) systems and cameras installed in front of the cage. Lidars and cameras were installed 1000 mm in front of the cage. **c** The four videos were assigned boxes and labels by Yolo. The face label parts were used for face identification, and the body portion was used for 3D tracking with Lidar centroid. The same detector was also used to detect grooming behavior. Finally, face identification and 3D tracking were linked to obtain the 3D trajectory of each marmoset and the 3D coordinates of the behavior.

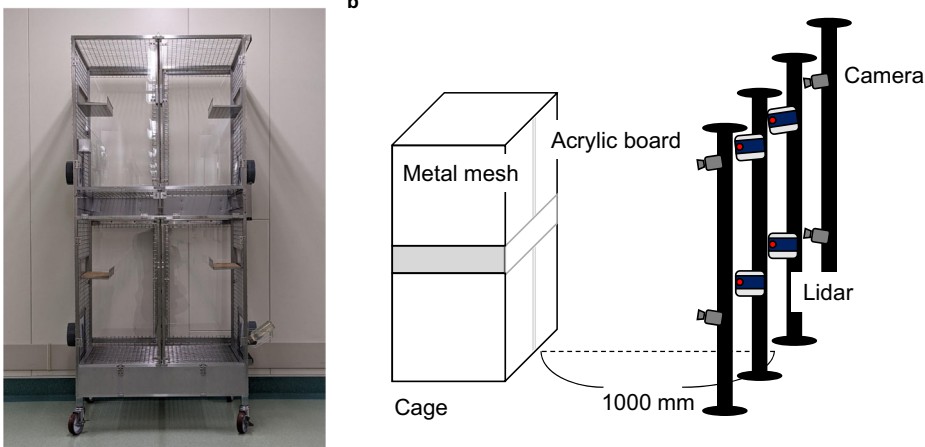

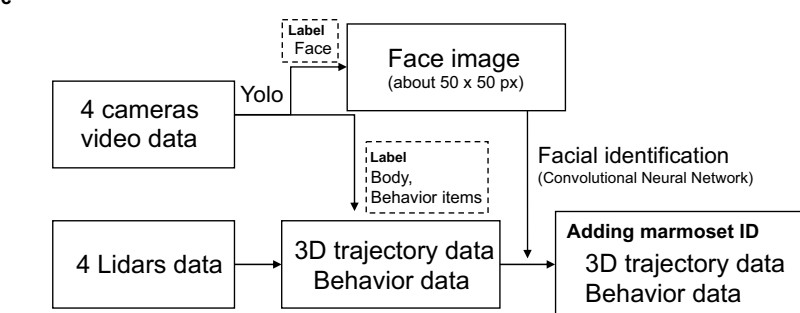

## Trajectory analysis using the 3D tracking system

Using the results of 3D trajectories from this 3D tracking system, we analyzed where and for how long each animal stayed in the cage during the 1-h period of the analyses (Fig. 4). This data indicated that marmoset A stayed on the lower floor of the cage for 88.8% (23,642/26,626 frames) of the 1 h, whereas both marmoset B and marmoset C stayed on the lower floor for less than 0.5% (128/25,597 and 12/24,234 frames) of the 1 h. In contrast, they stayed on middle floor for half of the 1 h and actively moved in various areas in the cage except the lower floor rest of the time (Fig. 4a–c, Table 3, Supplementary Fig. 1). Furthermore, the individual distances were analyzed using the location data (Fig. 4d). During this time, the distance between marmoset C and marmoset B was less than 0.5 m 25.9% (5044/19,501 frames) of the total time. However, a distance of less than 0.5 m between marmoset A-marmoset C and marmoset A-marmoset B was detected 1.0% (203/20,409 frames) and 2.2% (475/21,599 frames) of the time, respectively. Conversely, the distance between marmoset B-marmoset C was more than 1 m 2.0% (391/19,501 frames) of the time, whereas marmoset A-marmoset C and marmoset A-marmoset B were separated by more than 1 m distance 20.0% (4081/20,409 frames) and 11.0% (2364/21,599 frames) of the time, respectively. In this family, marmoset B and marmoset C spent more time together than marmoset A.

The same analyses were performed on a different day during the daytime, 7:00 am–7:00 pm. After 6:00 pm, all marmosets moved to the bed, and clustered together and their faces were not observed; therefore, the data were excluded from the analysis. The detailed analyses indicated that, among the family members, marmoset C stayed on the lower floor of the cage the longest, unlike the results of the 1-hour analysis (Supplementary Fig. 2). The time spent on the lower floor of the cage between 7:00 am and 6:00 pm for each individual was 8.6% for marmoset A, 13.0% for marmoset B, and 23.5% for marmoset C. Throughout the day, marmoset C spent the most time on the middle floor (27.7%), marmoset A on the upper floor (29.7%), and marmoset B on the upper floor

(29.9%). Furthermore, locations of each animal were analyzed hourly for 12 h (Supplementary Figs. 2, 3). Marmoset A also spent most of his time in the upper area in the morning (9:00 am; 54%), but gradually changed his activity area to the middle area in the afternoon (1:00 pm; 51%) and the lower area in the evening (4:00 pm; 45%). Marmoset B rarely stayed in the middle area but moved to the upper section in the morning (9:00 am; 48%) and to the lower section in the afternoon (3:00 pm; 55%). Marmoset C moved most evenly throughout the day among the three in the cage. At night, all individuals moved to the upper bunk to their beds (Supplementary Fig. 2).

The distance between the marmoset pairs was also analyzed throughout the day. The results showed that marmoset A-marmoset B were closer than 0.5 m for 21.1% of the daytime. However, marmoset A-marmoset C and marmoset B-marmoset C were closer than 0.5 m for 19.6% and 16.7%, respectively. Additionally, the closest and farthest pairs per hour switched frequently throughout the day (Supplementary Fig. 5). Although a similar analysis was performed for 76 days, the maximum continuous analysis period for this family was 35 days because of a downtime period due to system maintenance (Supplementary Fig. 6a). In another marmoset family, this system ran for 4 months continuously and was able to obtain longitudinal data (Supplementary Fig. 6b).

## Detection of grooming behavior

In this system, behaviors could also be detected using video data from the two upper cameras in front of the cage (Supplementary Fig. 7a, b). Grooming behavior, one of the representative social behaviors of marmosets, excluding self-grooming behavior was detected. Training data on grooming behavior were obtained through observations of video data by skilled animal technicians. To increase the accuracy of grooming behavior detection, situations in which marmosets were close to each other but not grooming were rigorously classified by skilled animal technicians as non-grooming images and eliminated from the supervised images. The model

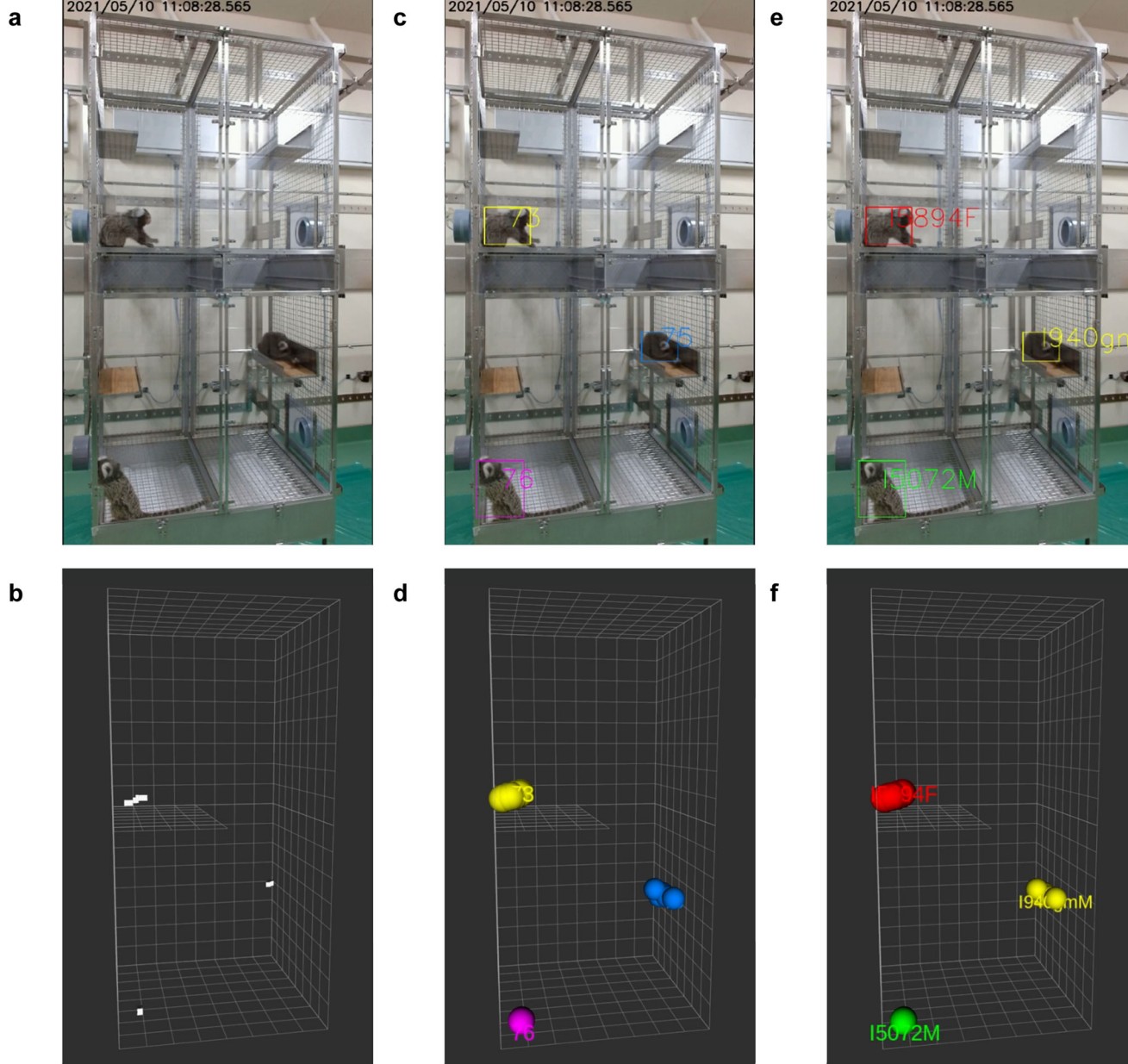

**Fig. 2 | 3D tracking of marmosets using Lidar and video tracking. a** Raw video frame. **b** Detected 3D coordinates of marmosets by Lidar. White point: centroid of marmosets; white box: virtual space of the cage. **c** Detected location of marmosets in video frame by Yolo. Rectangle: detected marmosets. **d** Calculated 3D coordinates of marmosets by video tracking, colored point: coordinate of marmoset. **e, f** Individual information added to (**c**) and (**d**) using face identification, green: marmoset A, red: marmoset B, yellow: marmoset C.

was trained by annotated data of the grooming positions in each video frame. Excluding annotation time, 880 h were spent on training. Creation of a csv file after processing of four movie files took 8 min. Face identification took 20 ms per process, and matching face identification to trajectory took 1 min per 10-min video.

The coordinates of the grooming location and individuals were detected by separately using the 3D tracking system, and determining which individual performed the grooming. Grooming behaviors were observed more often in the early morning (7:30–8:00 am), midday (12:30–1:30 pm), and evening (3:30–4:00 pm, 5:00 pm), and all marmosets performed grooming behaviors equally (Fig. 5a). All marmosets groomed at fixed locations (Fig. 5b), namely the two upper areas of the cage. These two locations were different from the sleeping area (Supplementary Fig. 3).

**Accuracy verification of the 3D tracking system**

To determine the accuracy of the 3D tracking system, marmosets were fitted with colored belly band to identify each animal, and individual tracking was conducted for 60 min. Supplementary Movie 4 shows a portion of this experiment and confirms that the system could track the marmosets accurately even when they crossed. Although the marmoset cage had a swinging hammock and balls as enrichment (Supplementary Movie 5), these movements did not affect the tracking accuracy of the marmosets when using a combination of video tracking and Lidar. During this experiment, the marmoset IDs switched to different IDs nine times for more than 5 s, but seven of the nine ID switching events were resolved within 15 s. The maximum time length for accurate individual tracking was 19 min. Incorrect marmoset IDs of longer than 5 s were exhibited over 4.6% of the total 60 min.

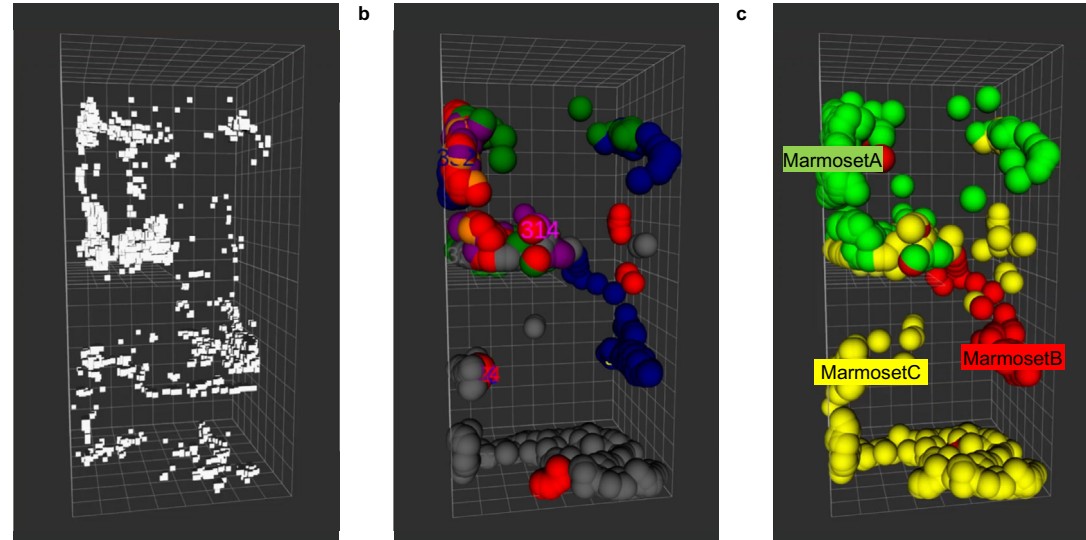

**Fig. 3 | 3D position trajectory of a marmoset over a period of 5 min. a** Marmoset trajectories analyzed by Lidar. White points indicate centroid of marmosets. **b** Lidar and video tracking. Different colors were assigned to each tracking serial number that numbered the background. **c** Lidar and video trajectory tracking combined with face recognition. Green: marmoset A; red: marmoset B; yellow: marmoset C.

The accuracy of the 3D trajectory data analysis in detecting the positions of animals in cages was confirmed by comparing it with visual observation data (Table 3, Supplementary Fig. 2). For example, in the visual observations of an hour, the position of the animals was on the upper wooden bed 9.4% of the time, on the middle floor 63.8%, on the lower wooden bed 25.1%, on the lower floor 0%, and in the other areas 1.7% of the time. Similarly, the system-analyzed data showed that the position of the animals on the upper wooden bed was 16.5%, middle floor was 52.6%, lower wooden bed was 29.1%, lower floor was 0.1%, and other areas was 1.7%. A strong significant correlation was found between the visual observation and system data ($R^2 = 0.988$, $P < 0.01$; Fig. 6a).

The accuracy of grooming behavior detection by the machine was compared with that of visual observation for 1 h (Fig. 6b). Each 30-s video was considered as one unit. Twenty-four grooming units were visually detected, and 21 units were detected by the machine. Among these, 19 units were detected by visual observation and machine (recall: 79.1%). In the system, 19 of 21 units were confirmed as actual grooming behaviors (precision: 90.5%). The five missed units were difficult to detect because of the placement of animals behind an object: four had marmosets grooming behind the feeder, and one had marmosets grooming in the blind spot of the bed. Conversely, two units were detected by the machine but not by direct observation. During these events, the marmoset nuzzled its face into the other's tail or placed its hand on the other's shoulder. These actions seemed to be part of grooming behavior; however, observers did not judge them as grooming behaviors because they were not typical grooming actions. In 1 h, there were ten units during which the distances between marmosets were close, and false positives were detected in only one unit, with a duration of 30 s (recall: 90%, Fig. 6b). The system also worked well in higher interaction situations, such as grooming. A grooming situation is shown in Supplementary Movie 6. The two animals were very close to each other and the animal ID switched frequently, but both were involved in grooming. After grooming, accurate individual identification was resumed as soon as the face was visible, as shown in Supplementary Movie 4.

## Discussion

In this study, we constructed the FulMAI system, an automatic 3D tracking system for multiple marmosets under free-moving conditions, using an approach that combines Lidar, video tracking, and deep learning face recognition. These are all existing technologies, but this is the first study combining them for 3D tracking of multiple unlabeled nonhuman primates with relatively rapid movements in free-moving conditions. The FulMAI system allowed us to analyze the natural behavior of individuals in a family for a month. Lidar and Yolo, the object detection algorithm, were fast programs that could be run constantly for a month with little delay. In addition, Yolo's light memory usage and easy integration with other systems will allow us to obtain more long-term and diverse data through well-planned operations in the future. FulMAI is expected to record and analyze longitudinal marmoset behavior and reveal what behavioral changes occur within the same individual before and after life events.

In preliminary studies, we used metal mesh doors, but these reflected the laser, and the marmosets' location could not be detected. The cage doors were changed to acrylic panels to make them suitable for the analysis (Fig. 1). The cages can be divided into four compartments. As shown in Supplementary Fig. 1, the system accurately tracked each animal even though the cage was divided. Therefore, the cage size can be changed to the extent that it fits within the camera's view. In contrast, thick poles should be avoided in front of the cage because they increase blind space for both the camera and the Lidar.

Although either laser or video data can be used as the primary data for this system, we used video tracking as a base for detecting marmoset positions and then completed the missing information using Lidar, thereby compensating for the lost marmoset trajectory detections by video and false recognitions by Lidar. Owing to its detection speed and resolution, Lidar is suitable for application to fast-moving objects. However, it falsely recognized objects that were similar in size to a marmoset, such as a marmoset tail, feeder boxes, or colored balls. On the other hand, a hammock in the family cage, which was included as an environmental enrichment and was not firmly fixed to the cage, was not mis-detected by either Lidar or video because of the differences in size and shape from those of the marmoset, and it did not affect the tracking of the marmosets. Video tracking does not have this issue because it uses color image data to identify objects (Supplementary Movie 5). However, video tracking lost detection when marmoset motion was quicker than the camera's shutter speed or when the marmosets were behind obstacles (Supplementary Movie 2, 3). By compensating for the respective shortcomings of Lidar and video tracking and using each to complement the trajectories of the other, a tracking system for small, high-speed objects in three dimensions was realized. In addition, because both Lidar and video tracking can be processed at high speed without a time lag, data is not accumulated for subsequent analysis, which is favorable for longitudinal analysis. Indeed, the 3D trajectories and grooming behaviors of

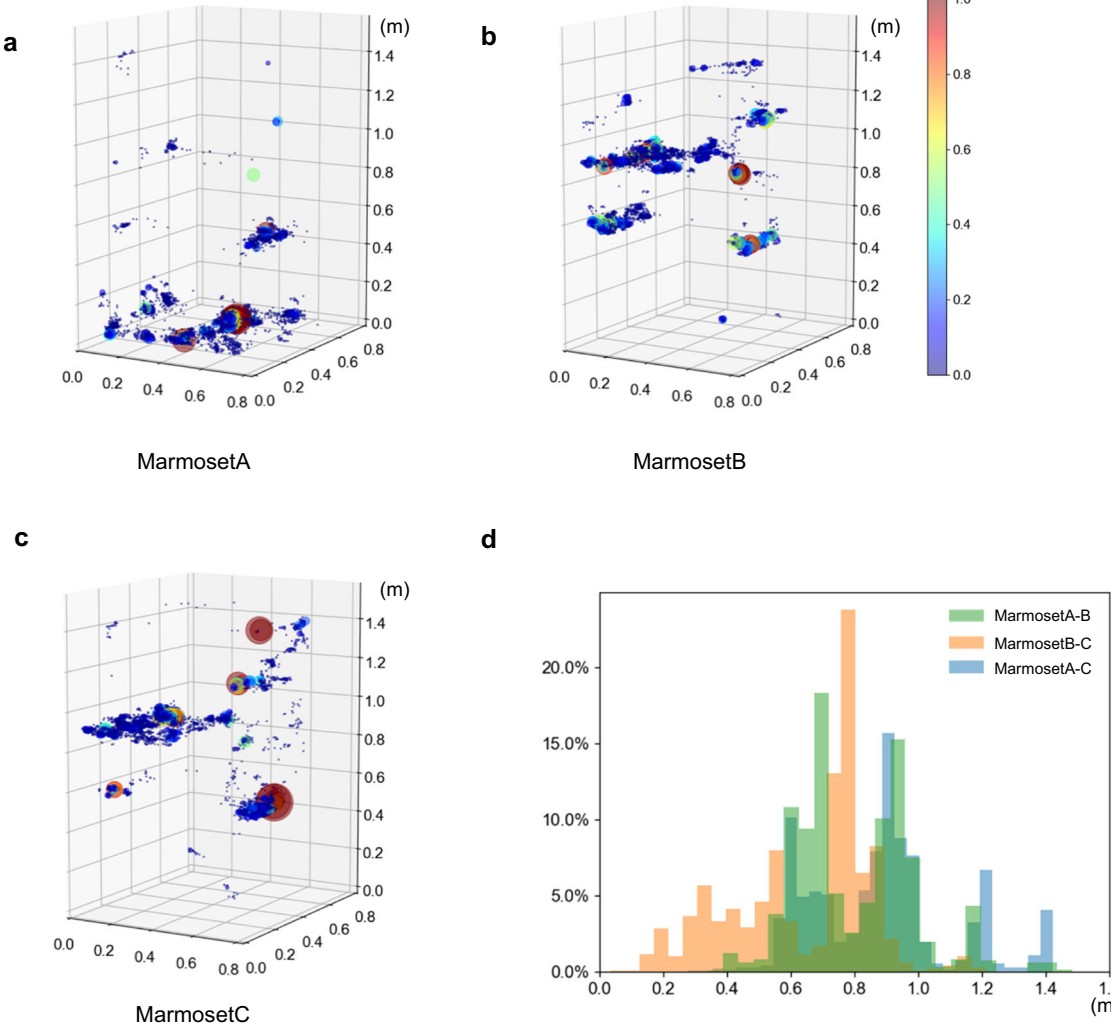

**Fig. 4 | Place preference of marmosets over a period of 1 h.** Color map of the time spent by marmosets in the cage for 1 h (**a–c**). Locations where marmosets spent more than 1% of 1 h are shown in red, and others are shown as color bars. **a** marmoset A, (**b**) marmoset B, (**c**) marmoset C. **d** Histogram of the distance between each family member. Blue bars indicate the individual distances between marmoset A and marmoset C, orange bars indicate the individual distances between marmoset B and marmoset C, and green bars indicate the individual distances between marmoset A and marmoset B.

three marmosets could be continuously analyzed in their home cage for over 1 month without data accumulation (Supplementary Fig. 6a). The animals were healthy throughout the experiment and had no apparent dysfunction caused by Lidar, proving the safety of our system. Although a similar 3D tracking system using Kinect or cameras has been reported for various animal species, including marmosets[1,11,14,15,21,22], none of the existing multiple 3D tracking systems have achieved such a continuous analysis for a month. Supplementary Movie 1, 4, and 5 show the same marmoset analysis data from the continuous FulMAI data at 5-month intervals. Yolo was trained only the first time during this period, and continuously obtained 3D trancking data without any ploblems. In addition, tracking information could be obtained for several consecutive months (Supplementary Fig. 6). Therefore, Yolo and Lidar are thought that no maintenance is required. On the other hand, face identification required monthly training. FulMAI would be able to analyze longitudinal behavioral changes over each marmoset's lifetime in the family. One of the advantages of FulMAI is the flexibility in terms of cage size and structure as long as acrylic panels are located in front of the camera and Lidar, and it can be applied to a wide variety of animal species. This system would also be useful for other animal species that move as quickly as marmosets in 3D, such as tamarins, free-swimming fish, and quick-flying birds and bats.

Our system successfully tracked multiple marker-less marmosets for the first time, which is another advantage of our system over existing systems. The color marking causes animal stress because it requires capture or anesthesia, and periodic hair dyeing requires time and effort, which is unsuitable for analyzing marmosets' behaviors in the long-term. The face identification of marmosets, which we were able to achieve for the first time, can also be undertaken independently and can be applied to other motion analysis systems, such as DeepBhvTracking[18].

When marmosets are kept as a family, they repeat the behavior of huddling and then suddenly dispersing several times a day. By this characteristic behavior, individual traceability is lost from the trajectory. Therefore, we combined our system with facial recognition, using deep learning to detect and track each animal in a family group, since the most distinctive feature of the marmoset's body is its face. Usually marmoset individual identification is performed using microchips, hair clippings, hair dyes, or necklaces[23]. Microchips and necklaces are unsuitable for MRI analysis, hair clippings and hair dyes require periodic maintenance, and necklaces pose the risk of tightening around the neck as they grow or of getting caught. In one part of this study, the marmosets were fitted with colored belly band for visual tracking, but marmoset B seemed uncomfortable with the band and was constantly fidgeting with it. Thus, artificial

**Table 3 | Comparison of visual observations and the system analysis of time spent in specific locations in the cage for an hour**

| | Marmoset A | | Marmoset B | | Marmoset C | |
|---|---|---|---|---|---|---|
| | Human | System | Human | System | Human | System |
| Upper bed | 0.4% | 0.6% | 10.7% | 16.5% | 9.4% | 16.5% |
| Middle floor | 0.3% | 0.6% | 38.8% | 52.6% | 63.8% | 52.6% |
| Lower bed | 8.8% | 8.9% | 41.8% | 29.1% | 25.1% | 29.1% |
| Lower floor | 90.3% | 88.8% | 0.5% | 0.5% | 0.0% | 0.1% |
| Other | 0.2% | 1.11% | 8.2% | 8.2% | 1.7% | 1.7% |

markers have the risk of causing unexpected behavioral changes. Therefore, one of the goals of this study was to achieve marker-less identification of individuals.

The marmoset family used in this study had no blood relationships with other family members since the juveniles (marmoset C) were obtained by an embryo from another embryo donor pair and was transferred into the female's uterus (marmoset B; foster mother)[24,25]. Regarding facial identification, the most accurate results within the family were those for marmoset C (Table 2). Although marmoset C was beyond sexual maturity, younger animals have easily identifiable distinctive facial features. In addition, when using five animals of the same age (1–2 years old), the animals could be individually identified; therefore, highly accurate classification is possible by facial recognition even when the animals are of similar age (Table 1). Incorporating facial recognition results into the trajectory enabled longitudinal tracking, even for multiple animals in the same cage (Fig. 3). We examined the accuracy of marmoset tracking using this system by color labeling the marmosets and comparing this to visual tracking (Supplementary Movie 4, Table 3). This experiment confirmed that it is possible to track animals using facial recognition with 95% accuracy, without labeling marmosets or relying on color information.

In addition, using the 3D coordinates data, this system could calculate where each animal stayed in the cage, for how long it stayed (Fig. 4a–c, Table 3, Supplementary Figs. 2, 3, 4), and the distance between individuals (Fig. 4d, Supplementary Fig. 5), which suggests interactions between individuals. Although switching of the marmoset IDs was observed several times during the hour-long experiment, results of visual observations and system analysis for where each marmoset existed in terms of 3D coordinates in the cage, and how long each marmoset spent in that area were highly correlated ($R^2 = 0.988$, Fig. 6a, Supplementary Fig. 2). This indicated that the slight ID switching did not affect the experimental results, and our tracking system was sufficiently accurate for automatically analyzing marmosets' 3D coordinates in the cage.

Marmosets often change their preferred position and the individuals that they spend time with during the day. For example, the marmoset A preferred the upper area in the morning, but in the afternoon, it preferred the lower bed. This result was not apparent in the 1-h analysis compared with observations. Furthermore, 1-h and 12-h analyses of position in the cage and distance between marmoset individuals showed different results. For example, a 1-h analysis showed that marmoset A preferred the cage's lower floor and remained there 80% of the analysis time (Table 3). However, a 12-h analysis indicated that marmoset C spent the longest time (23.5%) on the lower floor (Supplementary Fig. 3, 4). The analyses of the distance between individuals showed that marmoset B-marmoset C were within 0.5 m for 25.9% of the 1-h analysis, but the 12-h analysis indicated that the marmoset A-marmoset B remained closer than marmoset B-marmoset C or marmoset A-marmoset B. These results suggest that the results based on human observation for a limited time and those based on machine observation throughout the day may differ. Indeed, when we directly observed this family, marmoset A and marmoset B were usually far apart, but a 12-h analysis suggests that the marmoset A and marmoset B spent time close together. Therefore, it should also be considered that people's presence alters

marmoset behavior. To accurately understand marmoset behavior, it is necessary for the behavior analysis system to constantly analyze a marmoset's life with the same evaluation criteria. Our 3D tracking system can continuously analyze animal positions and distances between individuals.

In addition, some marmoset facilities, to provide more space or different environments for marmosets, occasionally connect adjacent marmosets' cages[26]. In this study, an extra cage was connected to the home cage. When a marmoset went off-screen due to movement to the connected cage, the tracking of the marmoset was temporarily interrupted. However, other marmosets continued to be tracked, and no other individual was tracked instead of the missing marmoset (Supplementary Movie 7). Tracking was resumed when the marmoset returned to the cage.

Grooming, a typical social behavior, was detected through deep learning using video analysis. The detector was Yolo, the same one used for the 3D tracking. Comparing detection of the grooming behavior using visual observation by humans and the system showed that grooming behaviors were detected with high precision (90.5%) and recall (79.1%; Fig. 6b). In some cases, the system identified behaviors just before or after the grooming as grooming behaviors. The deep learning system may have detected certain behaviors that might be unique to periods before and after grooming. However, in other cases, grooming behavior could not be detected by the deep learning system because the groomers were located in the camera's blind spot. Therefore, the training dataset must be further updated to detect behavior behind obstacles. Combining the 3D tracking with the grooming behavior detection by deep learning, this system could reveal when and where each marmoset displayed grooming behavior using accurate quantification, and the social relationships among the animals (Fig. 5 a, b). The 12-h grooming behavior analysis in this study indicated that grooming behavior was found to occur at specific locations: upper beds and floor, and at specific times: early morning, midday, and evening. This finding was obtained exclusively using automatic behavior analysis. While there are existing methods for automatically detecting the behavior of a single animal[27], for the first time, our system was able to directly detect the behavior of multiple animals in captivity and identify the individuals and their locations. We aim to identify a variety of behaviors for analyzing lifelong behavior changes. To find unpredictable unknown behavior, clustering by unsupervised learning would be necessary. For unsupervised learning, it is necessary to analyze the videos of each individual or to create a skeletal model of each marmoset and analyze the movement of each key point. Our study is expected to further develop in the future because of its ability to detect accurate 3D location linked to each individual. For more flexible behavior extraction, this system was made of Robot operating system (ROS) and is extensible to incorporate other Python-based projects, such as DeepLabCut, DeepEthogram[28], and Tweetynet[29]. Because all animal activities are recorded in the FulMAI, each animal behavior in this home cage can be analyzed retrospectively in detail. By comparing with other data, such as imaging, blood biochemistry, or biomarker data, it is also possible to analyze the behavior of model animals from multiple perspectives without overlooking changes in behavior associated with the life events of brain function changes or the onset of disease.

Moreover, Lidar can track the animal locations day and night; therefore, their nocturnal behavior can be observed to obtain primary data. Our preliminary data showed that the combination with a Lidar and starlight camera can capture details of nighttime behavior in a dark room (Supplementary Movie 8). Sleep disorders have recently been reported as predictors of Alzheimer's disease and Parkinson's disease[30,31]. Recently, we have established two kinds of Alzheimer's disease models[25] and expect to observe sleep disorders with this Lidar and starlight camera system.

Recently Calapai et al. reported a touch screen-based cognitive auditory analysis system that allows marmosets to perform tasks in a free-moving environment in a paired home cage[32]. By incorporating such a freely operable cognitive function task device into our system, it is possible to provide a behavior analysis system that comprehensively and continuously analyzes the marmoset's natural behavior, activity, and cognitive functions without inducing stress.

**a**

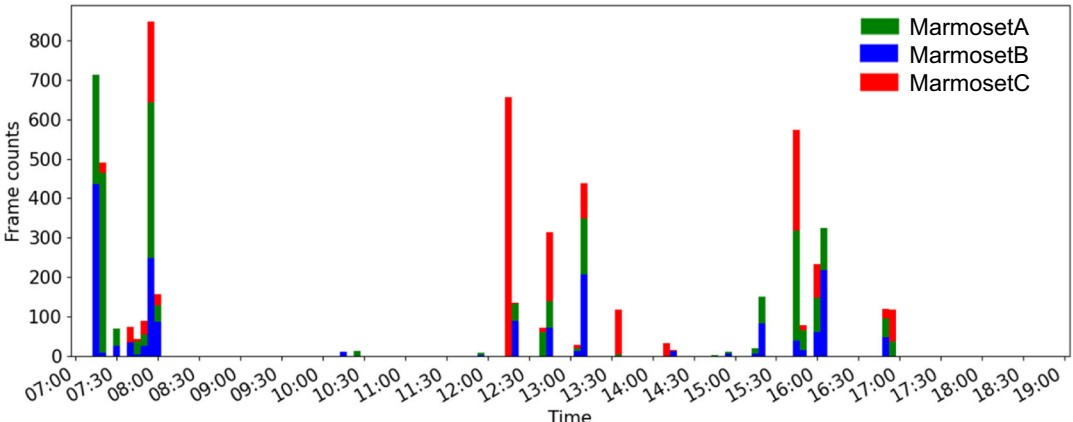

**b**

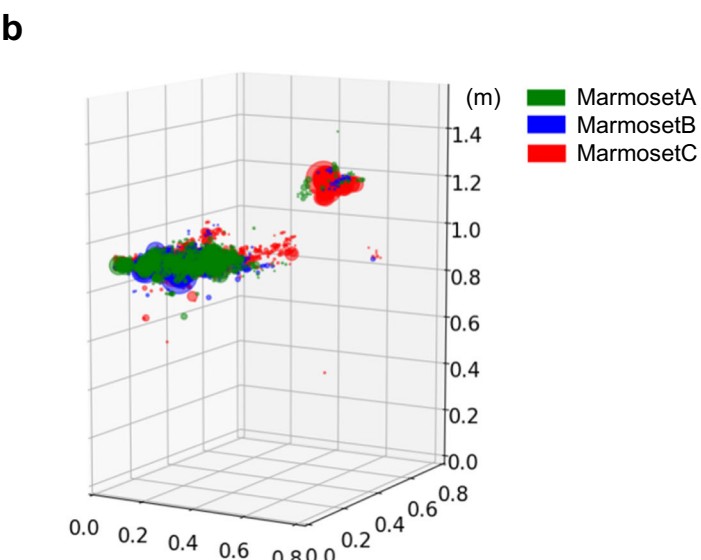

**Fig. 5 | Detection of grooming behavior for a day. a** Number of grooming behaviors detected per day. Bars show the number of frames detected each 5 min. **b** The 3D coordinates where the grooming behavior of each individual was detected are shown. Green indicates marmoset A, blue indicates marmoset B and red indicates marmoset C.

In conclusion, the combination of Lidar, video tracking, and facial recognition enabled successful simultaneous tracking of each marmoset in a family without artificial labeling. Using data from the 3D tracking system we developed, we could measure 3D coordinates, movement trajectories, the quantity of activity, position in the cage, when, and for how long each animal stayed, distances between individuals, and time spent in grooming behavior. These measurements will allow automated analysis of activity and behavior under free-moving conditions throughout the marmoset's lifetime, enabling quantitative analysis of "when, where, and what" each animal was doing and capturing behavior changes in development, growth, aging, and disease onset or progress.

## Materials and methods
### Hardware setup
The home cage was made of acrylic panels on the front and back, and all food dishes, water jugs, and bedding inside the cage were fixed to a wire mesh on the side to prevent them from moving. This cage was designed to conduct behavioral analysis in a marmoset's home cage (Fig. 1a). Four poles

were set up in front of the cage at 1000 mm, and four Lidars (VLP-16, Velodyne Lidar, San Jose, CA, USA) and four cameras (C920, Logitech, Lausanne, Switzerland) were fixed to the poles (Fig. 1b). The Lidar utilizes a 903 nm infrared laser light, which is eye-safe (Class I laser) and a reported range detection of up to 100 meters. The Lidar was controlled by motion measurement software, the path traced, and velocity of the marmosets recorded (Hitachi, Ltd, Tokyo, Japan.). The lower Lidar was set at 0.9 m height and the upper Lidar at 1.9 m height. The distance between the left and right Lidar devices was 0.3 m.

The laser beam of the Lidar was irradiated radially and adjusted to cover the entire interior area of the cage. The recording parameters of the cameras were 1080p and 60 Hz, and all cameras were adjusted to show the entire cage (Supplementary Fig. 7). The positions of the cage, Lidar, and cameras were calibrated when the cage was moved, and the interiors were changed. Lidar calibration was used to acquire background information, including the cage interior without marmosets. Calibration of the camera was performed by detecting alco-markers at the four corners of the cage. The specifications of the PC used to operate and record the camera and Lidar

**a**

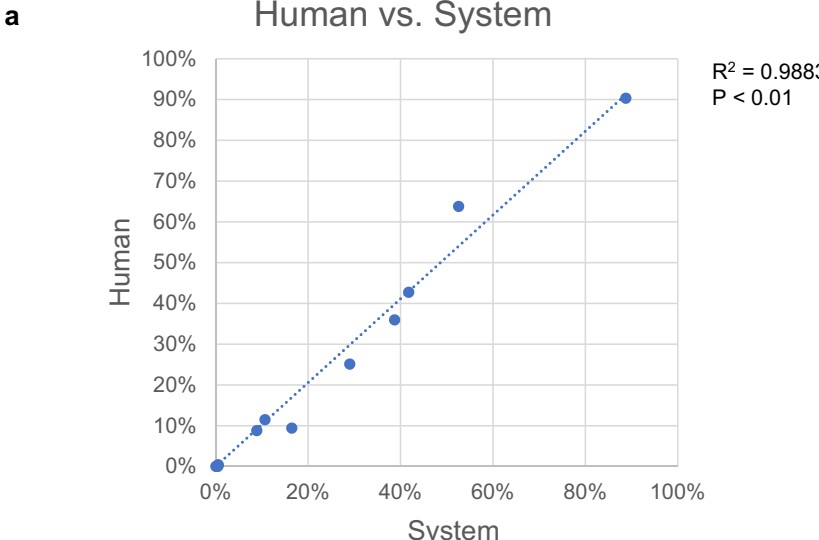

**b**

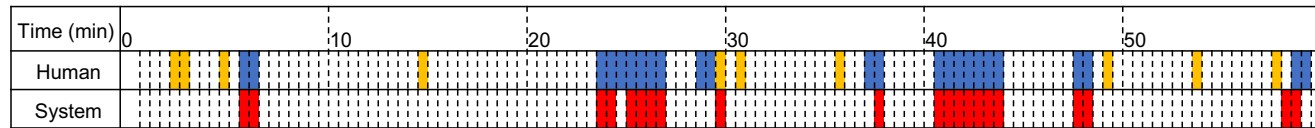

**Fig. 6 | Comparison of human visual observations and the system. a** The time the marmosets stayed in a particular location was compared between visual observations and those by the system. The marmosets stayed in four locations: upper bunk, middle floor, lower bunk, and lower floor. Dotted lines indicate regression lines.

**b** Each cell represented 30 s. Blue cells: Visual observation detected grooming at least once in 30 s. Yellow cells: Visual observation detected the close distance between two marmosets. Red cells: The system detected grooming by the intensity exceeding a threshold value.

were corei7, RTX1060, 32GB. PC specifications used for analysis were corei7, RTX3090, 64 GB.

### Ethical approval

This study was conducted at the animal facility of the Central Institute for Experimental Animals (CIEA), Kawasaki, Japan. The animal experimental protocol was reviewed by the Institutional Animal Care and Use Committee and approved (approval no. 17051) according to the Regulations for Animal Experiments in CIEA, based on the Guidelines for the Proper Conduct of Animal Experiments by the Science Council of Japan (2006).

### Animal facilities

The marmosets were kept in a family cage with dimensions of $820 \times 610 \times 1578$ mm. Although there was only one marmoset family in the room, they were able to vocally communicate with the marmosets in other rooms. All cages were equipped with a bunk, and the animal rooms were adjusted to 26–28 °C temperature, and 40–60% humidity with a 12:12 h light/dark cycle. The animals tested negative for *Salmonella* spp., *Shigella* spp., and *Yersinia pseudotuberculosis* on annual fecal examination. The study animals were inspected 9 months before the study.

### Animals

Each marmoset family was housed in a home cage. The observed family consisted of a male (marmoset A, 10 years old), female (marmoset B, 6 years old), and a juvenile (marmoset C, 1 year old). No artificial or non-artificial physical features were used for individual identification, such as hair dye, necklaces, piercings, or other visible signs, except during 3D tracking of marmosets during the color marker experiments. In conventional animal care, a microchip is implanted subcutaneously for individual identification. The individual marmoset C was born by embryo transfer to the mother's

uterus[25], and there was no blood relationship within the third degree of the family.

Five healthy marmosets ~1–2 years old, one male and four females, were used in the preliminary study to develop deep learning facial recognition (I774 1 year old, I6695 1 year old, I6708 1 year old, I6677 1 year old, and I893 2 years old). There were no artificial or visible physical characteristics, such as hair dye, necklaces, or piercings, and each marmoset was not related to any other marmoset within the third degree of kinship.

### Facial recognition

The cage ($W820 \times D610 \times H1578$ mm) was divided into four compartments to isolate each marmoset for obtaining training images by video camera. Each animal was housed in a separate cage, and videos were recorded from the cage front for 3–6 h. The front door of the cage was then replaced with clear acrylic panels. From these video frames, the face images were cropped only in the face area for the training images, and these data were augmented by rotation. Using 1400 facial images in each class for training, frontal images of the face were classified into four or five classes using the VGG19 model network. In the preliminary test, the five classes comprised five marmosets. For the family group, the four classes included those members and an unknown class to exclude images inappropriate for classification, and 1000 images were used for training.

Model training was performed by fine-tuning a pre-trained VGG19 model (learning rate 0.001, fixed number of layers 12, BATCH_SIZE 32)[33] (Supplementary Data 2). VGG19 is a network model proposed by the Visual Geometry Group at the University of Oxford in 2014 that obtained accurate classification performance on the ImageNet dataset. The pre-trained models used in this study were trained using the ImageNet dataset. For longitudinal analyses, training data collection and training were performed once a month to update the model.

## 3D tracking

Lidar and video tracking performed 3D tracking. A Lidar measures the distance to an object by irradiating a laser and analyzing the reflected laser. Marmoset detection using Lidar involves the following steps: background information removal, clustering, and selection of marmoset clusters[34]. Background removal was performed during calibration. The observed points were moving points belonging to objects whose spatial positions changed and static points from the background environment. Marmosets were then selected from among all the moving clusters. The Euclidean cluster extraction algorithm was used for clustering, with a minimum cluster size of S1 and maximum cluster size of S2 (S1: 100 mm, S2: 500 mm)[35]. The centroid of the cluster is regarded as the position of the marmoset for tracking. Three-dimensional video tracking was performed using four time-synchronized video cameras. The positions of the whole body and head of the marmoset were detected in each video frame using Yolo, an object detection algorithm, and the 3D position was calculated from each camera information[20]. The Yolo model was trained using 1164 images in which the whole body and head area were annotated (learning rate: 0.0003, momentum: 0.9, batch: 64). For tracking, the primary information used was the 3D video tracking coordinates, and the information was supplemented with Lidar coordinates when video coordinates were missing. When the primary video information was lost, the system searched for available Lidar coordinates close to the coordinates just before the location was lost and switched to Lidar tracking. When Lidar coordinates were available, continued tracking was undertaken using only Lidar. When the camera re-detected the coordinates, the tracking was switched back to the camera again. The switching was set to occur when the coordinate relationship between the camera and Lidar coordinates was fixed.

The program for linkage between video tracking and Lidar for individual information was produced by Hitachi, Ltd. When the marmoset coordinates were detected outside the cage, they were considered as missing values.

## Detecting grooming behavior

The software Yolo was used to detect grooming behavior. A total of 252 annotated image files in which grooming marmosets were surrounded by a minimum rectangle were prepared as training images, and the model was trained using the dataset. Grooming images were strictly categorized by skilled animal technicians to exclude situations in which marmosets did not groom. The trained model analyzed the top two camera view images and detected grooming behavior in each video frame. To improve the accuracy by removing noise, a Gaussian-weighted moving average of 50 s window and 5 s slide was applied to the frequency of grooming occurrence each time to create the probability distribution of grooming occurrence. The same process was performed for the left and right cameras and the calculated probability distributions were averaged. When the average value exceeded a certain threshold, grooming was considered to have occurred (Supplementary Data 2). To test the accuracy of this detection system, skilled breeding staff visually detected marmoset grooming in 1 h of video that was not used for training data. This visual result was used as the true label, and the accuracy and repeatability of grooming detection were calculated in 30 s increments of time. Because grooming was analyzed with two cameras, it was possible to link location and time. Grooming locations were calculated using the same method as used for video tracking. The grooming time was based on the video file.

## Statistical analysis

Pearson's correlation coefficient was calculated using GraphPad Prism version 9 (GraphPad Software, La Jolla, CA, USA).

## Reporting summary

Further information on research design is available in the Nature Portfolio Reporting Summary linked to this article.

## Data availability

All source data for the graphs are present in Supplementary Data 3. Additional data are available from the corresponding author on reasonable request.

## Materials availability

Information and requests for the program for linkage between video tracking and Lidar should be directed to and will be fulfilled by Norio Goda (norio.goda.ws@hitachi.com). Requests for other resources and information should be addressed to Erika Sasaki (esasaki@cieea.or.jp).

## Code availability

The readily tested code and data have been presented in the Supplemental Information (Supplementary Data 1, 2).

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

## Acknowledgements
This research was partially supported by the Brain Mapping by Integrated Neurotechnologies for Disease Studies (Brain/MINDS), "Study of developing neurodegenerative model marmosets and establishment of novel reproductive methodology (JP19dm0207065)" from the Japan Agency for Medical Research and Development (AMED) and Grant-in-Aid for Scientific Research (A) "Development of an Automated Behavioral Analysis System for Evaluation of Genetically Modified Disease Model Marmosets" JSPS KAKENHI Grant Numbers JP21H04756 to E.S. and Grant-in-Aid for Early-Career Scientists "Effects of transient alcohol exposure to the fetus in early pregnancy on brain development." JSPS KAKENHI Grant Number JP20K16908 to T.Y.

## Author contributions
T.Y. and E.S. conceived and designed the study. T.Y., N.G., J.K. and T.F. performed the setup of the measurement system. R.K., R.H., M.K., Y.H. and E.Y. performed the annotation of the training data. T.Y., O.K., Y.S., R.H., J.K. and T.F. processed the data, trained the models, and analyzed the results. T.Y. and R.K. performed the experiment. T.Y., W.K., K.S., N.G., T.I. and E.S. discussed and interpreted the results. T.Y., T.I. and E.S. wrote the manuscript. E.S. supervised the project. All authors commented on the manuscript.

## Competing interests
N.G. is an employee of Hitachi, Ltd. J.K. is an employee of Hitachi Solutions Technology Ltd. T.F. is an employee of Totec Amenity Limited. The remaining authors declare no competing interests.
