## [Peer review file · Communications Biology]

Reviewers' comments:

Reviewer #1 (Remarks to the Author):

Terumi Yurimoto et al. combined Lidar and deep learning algorithms to track 3D trajectories of animals and to recognize different marmosets under a free-moving home cage. This system provides a method to quantify the behavior of individual captive group animals and will facilitate the measurement of natural behavior, social behavior and parental behavior. This method is useful for tracking 3D trajectories of animals. However, it is very low efficient to recognize individual animals in a group and identify different kinds of behaviors. Below I provide some comments on how the authors could improve this very promising work, to facilitate communication of their rich data sets, and make it more digestible for the reader.

Majors:

1) The author uses both Lidar and deep learning algorithms to track 3D trajectories which will greatly improve the tracking accuracy of the animal's location. However, the Lidar system has a very low spatial resolution which cannot improve the identification of individual animals and their behaviors. So individual and behavior identification will only depend on the deep-learning algorithms -YOLO. With this method, the users must train multiple detectors for different behaviors and animals. So the identified efficiency is very low. Maybe it will be useful to compare with another method-deepLabCut that has been used in different animal models.

2)As described in the paper, this system had a good performance in homecage with acrylic panels, even when the animal moves fast, behind something, exits off-screen or it exists environment enrichments. How robust is the system in other free-moving conditions, like homecage surrounded with metal mesh (a more common type of cage) or three-chamber contexts (common paradigms for assessing social behavior)?

2) This method is a segregated system to obtain the 3D tracking of marmosets. What is the operational feasibility of this system? It is suggested to give the flow diagram of the system. Meanwhile, how much time is spent at each step including training, Yolo tracking and facial recognition?

3) Line 375-388: How to synchronous the recording information from four Lidars and four cameras is not clear.

4) Line 455-457: The description of detecting grooming behavior is not clear. Yolo v3 is an object detection algorithm to calculate the location of marmosets, how does it perform for selecting grooming behavior is not clear.

5) In their example videos and figures, most animals have very little behavior interaction. How about the accuracy and efficiency of this system in more interaction scenes?

Minors:

1) Add comparisons of the following literature in the introduction or discussion.

de Chaumont F, et al. Computerized video analysis of social interactions in mice. *Nat Methods*. 2012 Mar 4;9(4):410-7. doi: 10.1038/nmeth.1924. PMID: 22388289.

Sun GL, et al. DeepBhvTracking: A Novel Behavior Tracking Method for Laboratory Animals Based on Deep Learning. *Front Behav Neurosci*. 2021 Oct 28;15:750894. doi: 10.3389/fnbeh.2021.750894. PMID: 34776893; PMCID: PMC8581673.

2) Line 229-230: It is not the first report of simultaneous 3D tracking of multiple unmarked, freely moving nonhuman primates. See Lauer J, et, al., Multi-animal pose estimation, identification and

tracking with DeepLabCut. Nat Methods. 2022 Apr;19(4):496-504. doi: 10.1038/s41592-022-01443-0. Epub 2022 Apr 12. PMID: 35414125; PMCID: PMC9007739.

3) Line 651-652: This paper has been published online.

Reviewer #2 (Remarks to the Author):

In this study, the authors developed a method to track the 3D positions of multiple marmosets within the same cage without using any identification markers. A novel approach combined existing technology (Lider, Video tracking, and facial recognition). The authors demonstrated the developed method is capable of capturing the 3D position of multiple unmarked, freely moving marmosets in several recording sets using three animals. This is certainly a helpful technology to track the position of multiple animals moving at a higher speed, like the common marmoset.

Major comments.

1. At the end of the day, examples of only three marmosets are presented in this work. This is no problem for the technical paper to assess the performance of tracking methods. However, the authors implicate the result as the potential common profile of marmoset (e.g., lines 162-, 310-326), which rubbed me the wrong way. I recommend limiting the discussion to its technical aspect or increasing the number of animals if authors want to address the behavioral profile of the common marmoset.

2. The originality, advantage, and disadvantages of the proposed method among the comparable existing technology are not clearly stated. A number of key methods to capture freely moving animals have already been reported very recently that include MoSeq, maDLC, SimBA, B-SoiD, LEAP, SLEAP, MARS, DANNCE, FreiPose, not only DLC and OpenMonkeyStudio (discussed). Many of them are either capable of tracking multiple animals or 3D tracking or both. These are based on either supervised or unsupervised learning. Please provide a clear description of the developed method in the sense of the originality among these methods in the introduction and the advantage and disadvantages (limitations) in the discussion.

3. Specifically, what is the advantage against the recently published method by Lauer et al. (ref#30; which is now published in Nature Method)? This is briefly discussed (line 263-), but insufficient to judge the priority of the proposed method. The authors refer to the cost of the multiple high-speed cameras, but this is the trade-off to the cost of introducing the Lidar system (Liders, PCs, Software, etc.). We can estimate the cost for the cameras, but not the latter. Please clarify.

4. I am a bit confused about the process of assessing the accuracy of the system using grooming. I agree with the point that grooming is optimal behavior for assessing accuracy because the proposed system has a higher spatial resolution. The system seemed to detect the distance between individuals and decide if the two were close enough. Human observers probably judge the frame as the grooming by multiple factors, not only the distance. Then, the distance between marmosets is not a sufficient condition for grooming, and there could be a moment the distance is minimal but not deemed as grooming by the observer. Indeed, the authors briefly described this (line222-). However, I think this is not the accuracy of the system but the description of marmoset behavior (i.e., how frequently they groom each other when they stay closer?). The real test of the system is if the grooming and non-grooming behavior by the subtle difference of distance between animals. For example, if grooming or other (e.g., just touching the shoulder) can be dissociated by the system, then I think the system is accurate enough to detect the grooming.

5. Authors seemed to develop this technology exclusively for the common marmoset. Still, I think this paper will attract more general readers if it could be applied to other species. How advantageous of this system (compared to other systems) in its application to other animals?

6. Related to this point, a large part of the introduction and discussion is about the advantage of the markerless motion capture system in general, not specifically for the developed system per se, for the marmoset. Please limit discussion specifically about (dis)advantage of the developed system over the other system(e.g. see comment 3).

7. Similarly, the author's argument may be biased toward the potential analysis of the disease model. I think the combination of face recognition and Lider could be applicable for general behavioral analysis (as demonstrated in this work), not specifically disease models. Plus, the

supervised learning algorithm using this paper cannot detect the unpredictable behavioral phenotype of the marmoset disease model. Therefore, to make this paper more focused, I would recommend being more modest about this (e.g., Lines 60-68; 355-372.)

Minor comments,

1. Please provide detailed information as to the installation of LIDER (e.g., the distance between, the height from the floor) and the required specification of the PC system, and the extent of error of measurement.
2. If authors will refer to the usage of this system in the dark (nighttime), please describe if the proposed system could be applicable for that purpose.
3. Line 35-59. Introduction. The first two sentences can be eliminated because this is a technical paper about motion capturing.
4. Supplementary video is too large in size. It took minutes to download. It would be helpful to edit and extract the specific timeline.

Point by point responses to the Reviewers' Comments:

Reviewers' comments:

Reviewer #1 (Remarks to the Author):

Terumi Yurimoto et al. combined Lidar and deep learning algorithms to track 3D trajectories of animals and to recognize different marmosets under a free-moving home cage. This system provides a method to quantify the behavior of individual captive group animals and will facilitate the measurement of natural behavior, social behavior and parental behavior. This method is useful for tracking 3D trajectories of animals. However, it is very low efficient to recognize individual animals in a group and identify different kinds of behaviors. Below I provide some comments on how the authors could improve this very promising work, to facilitate communication of their rich data sets, and make it more digestible for the reader.

Thank you for your suggestions and comments. We have responded to each comment below.

Majors:

1) The author uses both Lidar and deep learning algorithms to track 3D trajectories which will greatly improve the tracking accuracy of the animal's location. However, the Lidar system has a very low spatial resolution which cannot improve the identification of individual animals and their behaviors. So individual and behavior identification will only depend on the deep-learning algorithms -YOLO. With this method, the users must train multiple detectors for different behaviors and animals. So the identified efficiency is very low. Maybe it will be useful to compare with another method-deepLabCut that has been used in different animal models.

Thank you for your comment. We agree with your comments about the spatial resolution of Lidar. Although the resolution of Lidar was high enough to recognize the shape of the marmoset's body and tail (please refer to Fig. A), a higher resolution was needed to identify individuals and their behavior. Therefore, as written in our previous manuscript, we used video data to detect marmoset positions and then completed the missing information using Lidar. The Lidar was used for detecting animals' location and 3D trajectory by the centroid of the point cloud (P7, line 119-122).

Our system aims to analyze and record the entire life of marmosets, which have a lifespan of approximately fifteen years and need real-time operation. Therefore, we first chose Yolo, which we knew to be lightweight and work, Yolo, and adapted it to our system. We used one Yolo detector to find marmoset body, face, and grooming behavior. Another advantage of Yolo is that this system was made of a Robot Operating System (ROS), and with its extra memory space, it is extensible to incorporate other Python-based projects, such as DeepLabCut (DLC), DeepEthogram, and Tweetynet. These additional systems are used to analyze the accumulated data in more detail on demand. We are considering using the extra memory to detect various behaviors other than grooming. As you pointed out, we agree that Yolo is unsuitable for unsupervised learning, and DLC may be suitable for detecting unexpected new behaviors.

On the other hand, the annotation work of DLC requires patience, so we believe there is a trade-off between efficiency for detection and preparation. We were also interested in the DLC, and we integrated the DLC into our system. However, its real-time operation was limited to 1 frame per second for integrating into our FulMAI system, which was unsuitable for real-time video data analysis. Therefore, we chose a smoother operation method, Yolo.

Fig. A. Two marmosets were detected by Lidar in a cage. The pink and yellow dots indicate marmoset bodies and the green dots indicate the cage. Light blue points are the centroids of marmoset, and red points are the coordinates.

Original manuscript:

The 3D coordinate information of marmosets was obtained using video tracking and Lidar (Fig. 2a-f, and Mov. S1).

Revised manuscript: Page 7, lines 119-122

The 3D coordinate information of marmosets was obtained using video tracking and Lidar (Fig. 2a-f, and Mov. S1). While the spatial resolution of Lidar was not enough to detect each marmoset and its behavior, it was sufficient to detect the body and tail. Therefore, the centroids of the point clouds were used as animal locations for 3D trajectory.

Revised manuscript: Page 13, lines 229-237,

In this study, we constructed the FulMAI system, a new automatic 3D tracking system for multiple marmosets under free-moving conditions, using a novel approach that combines Lidar, video tracking, and deep learning face recognition. These are all existing technologies, but this is the first study combining them for 3D tracking of multiple unlabeled nonhuman primates with relatively rapid movements in free-moving conditions. The FulMAI system allowed us to analyze the natural behavior of individuals in a family for a month. Lidar and Yolo, the object detection algorithm, were fast programs that could be run constantly for a month with little delay. In addition, Yolo's light memory usage and easy integration with other systems will allow us to obtain more long-term and diverse data through well-planned operations in the future.

Revised manuscript: Page 20, lines 354-360,

To find unpredictable unknown behavior, clustering by unsupervised learning would be necessary. For unsupervised learning, it is necessary to analyze the videos of each individual or to create a skeletal model of each marmoset and analyze the movement of each key point. Our study is expected to further develop in the future because of its ability to detect accurate 3D location linked to each individual. For more flexible behavior extraction, this system was made of Robot operating system (ROS) and is extensible to incorporate other Python-based projects, such as DeepLabCut, DeepEthogram³⁶, and Tweetynet³⁷.

2)As described in the paper, this system had a good performance in homepage with acrylic panels, even when the animal moves fast, behind something, exits off-screen or it exists environment enrichments. How robust is the system in other free-moving conditions, like homepage surrounded with metal mesh (a more common type of cage) or three-chamber contexts (common paradigms for assessing social behavior)?

Thank you for pointing this out. This system did not work well with metal mesh cages because the mesh frame reflected the laser, and the mesh frame inhibited the observation of animal behavior. Therefore, we employed acrylic panels for this system. Although a three-chamber context was not used in this study, our cage could be divided into four chambers, and it worked well in four separate sections (data not shown). Therefore, we believe that even in a three-chamber context, it can be implemented without any problem if acrylic panels are installed as the front doors. We have added the following sentences.

Revised manuscript: Page 14, lines 241-247,

In preliminary studies, we used metal mesh doors, but these reflected the laser, and the marmosets' location could not be detected. The cage doors were changed to acrylic panels to make them suitable for the analysis (Fig. 1). The cages can be divided into four compartments. As shown in Supplemental Fig. 1, the system accurately tracked each animal even though the cage was divided. Therefore, the cage size can be changed to the extent that it fits within the camera's view. In contrast, thick poles should be avoided in front of the cage because they increase blind space for both the camera and the Lidar.

3) This method is a segregated system to obtain the 3D tracking of marmosets. What is the operational feasibility of this system? It is suggested to give the flow diagram of the system. Meanwhile, how much time is spent at each step including training, Yolo tracking and facial recognition?

Thank you for your suggestion. A schema of our system has been added (Supplemental Fig. 6). We had conducted a test to run this system constantly and were able to run it continuously for one month with little delay. Therefore, we believe that longer term data can be obtained through planned management. Excluding annotation time, 880 hours were spent on training, including 30 minutes training time for facial recognition. Creation of a csv file after processing of four movie files took 8 minutes. Face identification took 20 msec per process, and matching face identification to trajectory took 1 minute per 10-minute video.

We added new sentences as follows.

Revised manuscript: Page 24, lines 430-439,

System workflow

The overview of this system is shown in Supplemental Fig. 7. The videos and Lidar data were processed on the same PC, and acquisition times were perfectly synchronized for at least one month. Yolo, an object recognition algorithm, detected the marmoset's face, body, actions, and environmental enrichments (ball and hammock) in each video frame. Body coordinates were combined with coordinates using Yolo and Lidar to calculate a short 3D trajectory. Furthermore, the cropped face images were identified as individuals using a convolutional neural network, and individual IDs were applied to the short-term 3D trajectories to achieve long-term 3D tracking of each individual. Behaviors were detected simultaneously with the same model of Yolo, and each animal's behavior was linked to the nearest marmoset ID.

Revised manuscript: Page 11, lines 184-186,

Excluding annotation time, 880 hours were spent on training. Creation of a csv file after processing of four movie files took 8 min. Face identification took 20 msec per process, and matching face identification to trajectory took 1 min per 10-min video.

4) Line 375-388: How to synchronous the recording information from four Lidars and four cameras is not clear.

We apologize for not having provided detailed information about the coordination of the video and Lidar data in the previous manuscript. We have added the following sentences in the revised manuscript in the Materials and Methods section as follows.

Original manuscript:

For video tracking, 3D coordinates were used as primary information and supplemented with Lidar coordinates when the primary information was missing.

Revised manuscript: Page 27, lines 468-475,

For tracking, the primary information used was the 3D video tracking coordinates, and the information was supplemented with Lidar coordinates when video coordinates were missing. When the primary video information was lost, the system searched for available Lidar coordinates close to the coordinates just before the location was lost and switched to Lidar tracking. When Lidar coordinates were available, continued tracking was undertaken using only Lidar. When the camera re-detected the coordinates, the tracking was switched back to the camera again. The switching was set to occur when the coordinate relationship between the camera and Lidar coordinates was fixed.

5) Line 455-457: The description of detecting grooming behavior is not clear. Yolo v3 is an object detection algorithm to calculate the location of marmosets, how does it perform for selecting grooming behavior is not clear.

Thank you for your suggestion. As shown in Supplemental Fig. 6, Yolo detected the face, body, and behavior. Because the training images of two marmosets together were strictly categorized as grooming or not grooming by skilled animal technicians, the trained Yolo model could distinguish between two marmosets and an individual. In addition, the calculation of Gaussian weighted moving average improved the accuracy of grooming detection. Yolo's performance in grooming behavior detection was compared to visual observation by skilled animal technicians, and the results are described in Fig. 6b. In this study, it was possible to distinguish between close contact and grooming. In Figure 6b, a yellow highlight has been added to indicate the time of close contact of two marmosets but not performing grooming behavior. Mov. S6 was also added as a video during grooming.

In response to your comment, we have revised our manuscript as follows:

Revised manuscript: Page 24, lines 430-439,

System workflow

The overview of this system is shown in Supplemental Fig. 7. The videos and Lidar data were processed on the same PC, and acquisition times were perfectly synchronized for at least one month. Yolo, an object recognition algorithm, detected the marmoset's face, body, actions, and environmental enrichments (ball and hammock) in each video frame. Body coordinates were combined with coordinates using Yolo and Lidar to calculate a short 3D trajectory. Furthermore, the cropped face images were identified as individuals using a convolutional neural network, and individual IDs were applied to the short-term 3D trajectories to achieve long-term 3D tracking of each individual. Behaviors were detected simultaneously with the same model of Yolo, and each animal's behavior was linked to the nearest marmoset ID.

Revised manuscript: Page 10, lines 180-183,

To increase the accuracy of grooming behavior detection, situations in which marmosets were close to each other but not grooming were rigorously classified by skilled animal technicians as non-grooming images and eliminated from the supervised images.

Revised manuscript: Page 13, lines 221-226,

In one hour, there were 10 units during which the distances between marmosets were close, and false positives were detected in only one unit, with a duration of 30 sec (recall: 90%, Fig. 6b). The system

also worked well in higher interaction situations, such as grooming. A grooming situation is shown in Mov. S6. The two animals were very close to each other and the animal ID switched frequently, but both were involved in grooming. After grooming, accurate individual identification was resumed as soon as the face was visible, as shown in Mov. S4.

Original manuscript: Fig. 6 Figure legend:

(a) Each cell represented 30 s. Blue cells: Visual observation detected grooming at least once in 30 s. Red cells: The system detected grooming by the intensity exceeding a threshold value.

Revised manuscript: Fig. 6 Figure legend: Page 32, lines 558-561,

(b) Each cell represented 30 s. Blue cells: Visual observation detected grooming at least once in 30 s. Yellow cells: Visual observation detected the close distance between two marmosets. Red cells: The system detected grooming by the intensity exceeding a threshold value.

6) In their example videos and figures, most animals have very little behavior interaction. How about the accuracy and efficiency of this system in more interaction scenes?

Thank you for your suggestion. The system also works in more interactive situations, such as during grooming behavior. We have added Mov. S6, which includes more interactive situations. The system could identify individual animals even if the animals were clustered together, although the animal IDs were switched frequently. There was no effect on accuracy, as the system would return to accurate tracking once marmosets separated (Mov. S4). In the revised Figure 6b, we have added yellow highlights, which indicate that the animals were close to each other but not grooming. In the one hour of observation, there were 10 units in which marmosets were close to each other, and false positives were detected only once (recall: 90%). Therefore, we think that close contact and grooming can be accurately separated.

As described above, we have added the following sentences.

Revised manuscript: Page 13, lines 221-226,

In one hour, there were 10 units during which the distances between marmosets were close, and false positives were detected in only one unit, with a duration of 30 sec (recall: 90%, Fig. 6b). The system also worked well in higher interaction situations, such as grooming. A grooming situation is shown in Mov. S6. The two animals were very close to each other and the animal ID switched frequently, but both were involved in grooming. After grooming, accurate individual identification was resumed as soon as the face was visible, as shown in Mov. S4.

Minors:

1) Add comparisons of the following literature in the introduction or discussion.

De Chaumont F, et al. Computerized video analysis of social interactions in mice. *Nat Methods*. 2012 Mar 4;9(4):410-7. Doi: 10.1038/nmeth.1924. PMID: 22388289.

Sun GL, et al. DeepBhvTracking: A Novel Behavior Tracking Method for Laboratory Animals Based on Deep Learning. *Front Behav Neurosci*. 2021 Oct 28;15:750894. Doi: 10.3389/fnbeh.2021.750894. PMID: 34776893; PMCID: PMC8581673.

Thank you for this suggestion. The Chaumont et al. reference has been added in the Introduction section (line 63, Ref. 25). The Sun GL et al. reference has been added in the Introduction and Discussion sections (line 66, 276, Ref. 27)

Revised manuscript: Page 4, lines 60-69,

The various systems used for analyzing animal behaviors under free-moving activities include systems suitable for three dimensions (3D) tracking and behavior classification (DANNCE¹⁸, FreiPose¹⁹, and MarmoDetecotor²⁰), for generating pose estimation (DeepLabCut, SLEAP²¹, MARS²², DANNCE,

FreiPose, and OpenMonkeyStudio¹), for analyzing behavior in home cages (DeepLabCut^{23, 24}, B-SoiD¹⁷, SLEAP, and MARS), for analyzing multiple animals (DeepLabCut, MiceProfiler²⁵, SLEAP, and OpenMonkeyStudio), for finding novel behavioral abnormalities by unsupervised clustering (MoSeq²⁶, B-SoiD, and FreiPose¹⁹), and for use across animal species (DeepLabCut, DeepBhvTracking²⁷, B-SoiD, and DANNCE). These systems have been generally developed for mice; only DeepLabCut, DeepBhvTracking, DANNCE, and MarmoDetector²⁰ have been adapted to marmosets, which move relatively quickly in 3D.

Revised manuscript: Page 16, lines 277-279,

The face identification of marmosets, which we were able to achieve for the first time, can also be undertaken independently and can be applied to other motion analysis systems, such as DeepBhvTracking²⁷.

2) Line 229-230: It is not the first report of simultaneous 3D tracking of multiple unmarked, freely moving nonhuman primates. See Lauer J, et, al., Multi-animal pose estimation, identification and tracking with DeepLabCut. Nat Methods. 2022 Apr;19(4):496-504. Doi: 10.1038/s41592-022-01443-0. Epub 2022 Apr 12. PMID: 35414125; PMCID: PMC9007739.

The original manuscript referred to DeepLabCut as being used to analyze two marmosets (original manuscript p500, lines 35-47, Ref. No. 23). However, the paper stated that one marmoset (for each pair) was dyed in blue on their ear tuft for animal identification (please refer to the materials and methods section of reference paper 23). Since our system used CNN for animal identification, we believe that our study is the first to track markerless marmosets.

3) Line 651-652: This paper has been published online.

Thank you for the pointing this out.

Reference No. 23 has been updated to the online publication.

Reviewer #2 (Remarks to the Author):

In this study, the authors developed a method to track the 3D positions of multiple marmosets within the same cage without using any identification markers. A novel approach combined existing technology (Lidar, Video tracking, and facial recognition). The authors demonstrated the developed method is capable of capturing the 3D position of multiple unmarked, freely moving marmosets in several recording sets using three animals. This is certainly a helpful technology to track the position of multiple animals moving at a higher speed, like the common marmoset.

Thank you for your suggestions and comments. We have responded to your comments below.

Major comments.

1. At the end of the day, examples of only three marmosets are presented in this work. This is no problem for the technical paper to assess the performance of tracking methods. However, the authors implicate the result as the potential common profile of marmoset (e.g., lines 162-, 310-326), which rubbed me the wrong way. I recommend limiting the discussion to its technical aspect or increasing the number of animals if authors want to address the behavioral profile of the common marmoset.

Thank you very much for your comment. We apologize for our unclear description. We intended to have a technical discussion and did not mean to discuss the general marmoset character based on the results from three marmoset individuals used in this experiment. The profile seen in this study was limited to the family. We have revised the description to be limited to family members. All the words “father,” “mother,” “male,” “female,” and “juvenile” have been changed to “I5072M/f (father),” “I5894F/m (mother),” and “I940M/j (juvenile)”. Since we think the relationship between each animal was important for understanding their behaviors. We have retained the words “father,” “mother,” and “offspring,” and gave them individual IDs.

Original manuscript:

All 3D tracking and behavioral tests were conducted using a single family consisting of three marmosets (male, female, and juvenile) in a cage.

Revised manuscript: Page 7, lines 116-118,

All 3D tracking and grooming behavior detection tests were conducted using data from one family consisting of three marmosets (I5072M/father (I5072M/f), I5894F/mother (I5894F/m), and I940M/juvenile (I940M/j)) in a cage.

Original manuscript:

This data indicated that the male (father) stayed on the lower floor of the cage for 80.2% of the one hour, while both the female (mother) and juvenile stayed on the lower floor for 0.4% of the one hour.

Revised manuscript: Page 8, lines 143-146,

This data indicated that I5072M/f stayed on the lower floor of the cage for 88.8% (23642/26626 frames) of the one hour, whereas both I5894F/m and I940M/j stayed on the lower floor for less than 0.5% (128/25597 and 12/24234 frames) of the one hour.

Original manuscript:

During this time, the distance between the juvenile and female was less than 0.5 m 25.9% of the total time. However, a distance of less than 0.5 m between juvenile-male and male-female was detected 1.0% and 2.2% of the time, respectively. Conversely, the juvenile-female distance was separated by more than 1 m 2.0% of the time, while the juvenile-male and male-female distance were separated by

more than 1 m 20.0% and 11.0% of the time, respectively. From these results, it appeared that the juvenile preferred to be closer to the female than to the male.

Revised manuscript: Page 9, lines 149-155,

During this time, the distance between I940M/j and I5894F/m was less than 0.5 m 25.9% (5044/19501 frames) of the total time. However, a distance of less than 0.5 m between I940M/j-I5072M/f and I5072M/f-I5894F/m was detected 1.0% (203/20409 frames) and 2.2% (475/21599 frames) of the time, respectively. Conversely, the distance between I940M/j-I5894F/m was more than 1 m 2.0% (391/19501 frames) of the time, whereas I940M/j-I5072M/f and I5072M/f-I5894F/m were separated by more than 1 m distance 20.0% (4081/20409 frames) and 11.0% (2364/21599 frames) of the time, respectively. In this family, I940M/j and I5894F/m spent more time together than I5072M/f.

Original manuscript:

The detailed analyses indicated that, among the family members, the juvenile stayed on the lower floor of the cage the longest, unlike the results of the 1-hour analysis (Supplementary Fig. 2). The time spent on the lower floor of the cage between 7:00 am and 6:00 pm for each individual was male 8.6%, female 13.0%, and juvenile 23.5%. Throughout the day, the juvenile spent the most time on the middle floor (27.7%), the male on the upper floor (29.7%), and the female on the upper floor (29.9%). Furthermore, locations of each animal were analyzed hourly for 12 hours (Supplementary Fig. 2, 3). The male also spent most of his time in the upper area in the morning (9:00 am; 54%), but gradually changed his activity area to the middle area in the afternoon (1:00 pm; 51%) and the lower area in the evening (4:00 pm; 45%). The female rarely stayed in the middle area but moved to the upper section in the morning (9:00 am; 48%) and to the lower section in the afternoon (3:00 pm; 55%). The juvenile moved most evenly throughout the day among the three in cage.

Revised manuscript: Page 10, lines 158-169,

The detailed analyses indicated that, among the family members, I940M/j stayed on the lower floor of the cage the longest, unlike the results of the 1-hour analysis (Supplementary Fig. 2). The time spent on the lower floor of the cage between 7:00 am and 6:00 pm for each individual was 8.6% for I5072M/f, 13.0% for I5894F/m, and 23.5% for I940M/j. Throughout the day, I940M/j spent the most time on the middle floor (27.7%), I5072M/f on the upper floor (29.7%), and I5894F/m on the upper floor (29.9%). Furthermore, locations of each animal were analyzed hourly for 12 hours (Supplementary Figs. 2, 3). I5072M/f also spent most of his time in the upper area in the morning (9:00 am; 54%), but gradually changed his activity area to the middle area in the afternoon (1:00 pm; 51%) and the lower area in the evening (4:00 pm; 45%). I5894F/m rarely stayed in the middle area but moved to the upper section in the morning (9:00 am; 48%) and to the lower section in the afternoon (3:00 pm; 55%). I940M/j moved most evenly throughout the day among the three in the cage.

Original manuscript:

The results showed the male and female were closer than 0.5 m for 21.1% of the daytime. However, the male and the juvenile, and female and juvenile were closer than 0.5 m for 19.6% and 16.7%, respectively.

Revised manuscript: Page 10, lines 171-173,

The results showed that I5072M/f-I5894F/m were closer than 0.5 m for 21.1% of the daytime. However, I5072M/f-I940M/j and I5894F/m-I940M/j were closer than 0.5 m for 19.6% and 16.7%, respectively.

Original manuscript:

The juvenile was born by embryo transfer to the mother's uterus, and there was no blood relationship within the third degree of the family.

Revised manuscript: Page 26, lines 423-424,

The individual I940M/j was born by embryo transfer to the mother's uterus¹⁰, and there was no blood relationship within the third degree of the family.

Original manuscript: Fig. 3 Figure legend

(c) Lidar and video trajectory tracking combined with face recognition. Green: male; red: female;

yellow: juvenile.

Revised manuscript: Fig. 3 Figure legend: Page 34, lines 542-543,

(c) Lidar and video trajectory tracking combined with face recognition. Green: I5072M/f; red: I5894F/m; yellow: I940M/j.

Original manuscript: Fig. 4 Figure legend

Color map of the time spent by marmosets in the cage for one hour (a-c). Locations where marmosets spent more than 1% of one hour are shown in red, and others are shown as color bars. (a) Male (I5072M), (b) female (I5894F), (c) juvenile (I940M). (d) Histogram of the distance between each family member. Blue bars indicate the individual distances between the juvenile (I940M) and the male (I5072M), orange bars indicate the individual distances between the juvenile and the female (I5894F), and green bars indicate the individual distance between the male and the female.

Revised manuscript: Fig. 4 Figure legend: Page 31, lines 545-550,

Color map of the time spent by marmosets in the cage for one hour (a-c). Locations where marmosets spent more than 1% of one hour are shown in red, and others are shown as color bars. (a) I5072M/f, (b) I5894F/m, (c) I940M/j. (d) Histogram of the distance between each family member. Blue bars indicate the individual distances between I940M/j and I5072M/f, orange bars indicate the individual distances between I940M/j and I5894F/m, and green bars indicate the individual distances between I5072M/f and I5894F/m.

Original manuscript: Fig. 5 Figure legend

(b) The 3D coordinates where the grooming behavior of each individual was detected are shown. Red indicates juvenile (I940M), green indicates male (I5072M), and blue indicates female (I5894F).

Revised manuscript: Fig. 5 Figure legend: Page 33, lines 553-554,

(b) The 3D coordinates where the grooming behavior of each individual was detected are shown. Red indicates I940M/j, green indicates I5072M/f, and blue indicates I5894F/m.

2. The originality, advantage, and disadvantages of the proposed method among the comparable existing technology are not clearly stated. A number of key methods to capture freely moving animals have already been reported very recently that include MoSeq, maDLC, SimBA, B-SoiD, LEAP, SLEAP, MARS, DANNCE, FreiPose, not only DLC and OpenMonkeyStudio (discussed). Many of them are either capable of tracking multiple animals or 3D tracking or both. These are based on either supervised or unsupervised learning. Please provide a clear description of the developed method in the sense of the originality among these methods in the introduction and the advantage and disadvantages (limitations) in the discussion.

Thank you for your valuable suggestion. We have added references and originality in the introduction of the revised manuscript. We think that the advantage of our system is that three marmosets can be tracked individually without marking and social behavior within the family can be continuously observed in a home cage, such as confirming details of grooming such as the location, timing, and the identity of the individuals grooming each other. Further, we think that the real-time analysis performance of our system is suitable for analyzing behavioral changes throughout the animal's life. The concept of recording and analyzing an animal's lifetime behavior is unique, and our system was designed to fit this concept. This is the only home cage system that allows retrospective and detailed reanalysis of model animals even after being found to have developed disease if kept in this cage. In addition, we included the advantages of applying the methods to other animal species in the discussion section. In the Discussion section, limitations regarding the detection of unknown behavior of this system were described. As you mentioned, we also think that unsupervised learning is necessary to find unknown behaviors, and we plan to combine this with pose estimation methods in the future. One of the advantages of our system was that the system could run with relatively low memory. This made it possible to integrate other systems, for example DeepLabCut, with the extra memory.

Original manuscript:

However, when a novel disease model is developed, it is unclear when and how behavioral changes

and social interactions will arise due to the disease. Even if multiple types of exhaustive behavioral task tests are conducted, it is not always possible to accurately capture behavioral changes through these tests. Further, it is often challenging to detect behavioral changes, such as increased appetite, hypoactivity, aggression, irritability, and apathy, using behavioral task tests. It is therefore important to clarify various behavioral changes, including changes in social interactions, in animal disease models based on their behavior during free activity. In mice, there are some systems for analyzing animal behavior during free-ranging activities under group-housed conditions.

Revised manuscript: Page 4, lines 56-59,

Automated behavior analysis in a home cage enable observations of more natural animal behavior^{14, 15, 16}. This analysis has the advantages of being able to ignore environmental condition, monitor social behavior in multiple animals, and analyze changes in social behavior over a relatively long period of time¹⁷.

Revised manuscript: Page 4, lines 59-68,

The various systems for analyzing animal behaviors under free-moving activities include systems suitable for three dimensions (3D) tracking and behavior classification (DANNCE¹⁸, FreiPose¹⁹, and MarmoDetecotor²⁰), for generating pose estimation (DeepLabCut, SLEAP²¹, MARS²², DANNCE, FreiPose, and OpenMonkeyStudio¹), for analyzing behavior in home cages (DeepLabCut^{23, 24}, B-SoiD¹⁷, SLEAP, and MARS), for analyzing multiple animals (DeepLabCut, MiceProfier²⁵, SLEAP, and OpenMonkeyStudio), for finding novel behavioral abnormalities by unsupervised clustering (MoSeq²⁶, B-SoiD, and FreiPose¹⁹), and for use across animal species (DeepLabCut, DeepBhvTracking²⁷, B-SoiD, and DANNCE). These systems have been developed for mice; therefore, only DeepLabCut, DeepBhvTracking, DANNCE, and MarmoDetector: ²⁰have been adapted to marmosets, which move quickly in 3D.

Revised manuscript: Page 5, lines 77-89,

In this study, we have developed a behavioral analysis system named FulMAI (Full Monitoring and Animal Identification), that can record and analyze each animal's behaviors living within multiple animals for their whole life. The concept of this system is to detect changes in behavior over the lifespan and understand the relation of changes in brain function. FulMAI is a 3D tracking system using cameras, light detection and ranging (Lidar) devices and deep learning to simultaneously track a marmoset family of three animals in one cage. Furthermore, to take advantage of stress-free behavior analysis in the home cage, we also developed a system for individual identification by facial recognition using deep learning. The FulMAI system is able to analyze where and for how long each animal stayed in the cage and the distance between individuals. Further, the time spent in social behavior is also an index for analyzing animal interactions. Therefore, as an example of social behavior detection, we also developed automatic grooming behaviors detection system using deep learning. The technology developed in this study would be applied in detecting important behavioral changes caused by development, aging, and various diseases in small nonhuman model primates.

Revised manuscript: Page 13, lines 228-236,

This study constructed the FulMAI system, a new automatic 3D tracking system for multiple marmosets under free-moving conditions, using a novel approach that combines Lidar, video tracking, and deep learning face recognition. These are all existing technologies, but this is the first study combining them for 3D tracking of multiple unlabeled nonhuman primates when they were rapid and free moving. The FulMAI system allowed us to analyze the natural behavior of individuals in a family for a month. Lidar and Yolo, the object detection algorithm, were fast programs that could be run constantly for a month with little delay. In addition, Yolo's light memory usage and easy integration with other systems will allow us to obtain more long-term and diverse data through well-planned operations.

Revised manuscript: Page 15, lines 259-263,

In addition, because both Lidar and video tracking can be processed at high speed without a time lag, no accumulation of data for subsequent analysis occurs, which is favorable for long-term analysis. Indeed, the 3D trajectories and grooming behaviors of three marmosets could be continuously analyzed in their home cage for over six months without data accumulation. The animals were healthy throughout the experiment, proving the safety of our system.

Revised manuscript: Page 15, lines 267-271,

One of the FulMAI's advantage is the flexibility in terms of cage size and structure as long as acrylic panels are located in front of the camera and Lidar, and it can be applied to a wide variety of animal species. This system would also be useful for other animal species that move as quickly as marmosets in 3D, such as tamarins, free-swimming fish, and quick-flying birds and bats.

Revised manuscript: Page 20, lines 346-360,

This finding was only obtained using automatic behavior analysis. While there are existing methods for automatically detecting the behavior of a single animal³⁵, our system was able to directly detect the behavior of multiple animals in captivity and identify the individuals and their locations for the first time. We plan to identify a variety of behaviors for analyzing lifelong behavior changes. To find unpredictable unknown behavior, clustering by unsupervised learning would be necessary. For unsupervised learning, it is necessary to analyze the videos of each individual or to create a skeletal model of each marmoset and analyze the movement of each key point. Our study is expected to be developed in the future because of its ability to detect accurate 3D location linked to each individual. For more flexible behavior extraction, this system was made of Robot operating system (ROS) and is extensible to incorporate other Python-based projects, such as DeepLabCut, DeepEthogram³⁶, and TweetyNet³⁷. Because all animals' activities are recorded in the FulMAI, each animal behaviors in this home cage can be analyzed retrospectively in detail. By comparing with other data, such as imaging, blood biochemistry, or biomarker data, it is also possible to analyze the behavior of model animals from multiple perspectives without overlooking changes in behavior associated with the life events of brain function changes or the onset of disease.

3. Specifically, what is the advantage against the recently published method by Lauer et al. (ref#30; which is now published in Nature Method)? This is briefly discussed (line 263-), but insufficient to judge the priority of the proposed method. The authors refer to the cost of the multiple high-speed cameras, but this is the trade-off to the cost of introducing the Lidar system (Lidars, PCs, Software, etc.). We can estimate the cost for the cameras, but not the latter. Please clarify.

We apologize that the hardware cost comparison was not appropriate; even a high-speed camera costs about \$1450, and a Lidar \$850. The high-resolution and high-speed cameras often used in DeepLabCut had limited recording time because of larger data and did not fit our goal of continuously recording over the animal's lifetime. When the marmosets make a rare pose like a somersault, only DeepLabCut was insufficient and occasionally failed to detect animal pose in our environment. In these cases, Lidar supplementation was useful. Moreover, Yolo's low annotation cost and lightweight operation and memory were advantages for us and could be combined with other deep learning tools (face identification and behavior analysis).

Therefore, our manuscript has been revised as follows.

Original manuscript: delete

DeepLabCut, is a powerful tool for estimating the posture of an animal using only a camera, but its accuracy depends on the shutter speed of the camera^{29, 32}.

Revised manuscript: Page 15, lines 254-259,

Video tracking does not have this issue because it uses color image data to identify objects (Mov. S5). However, video tracking lost detection when marmoset motion was quicker than the camera's shutter speed or when the marmosets were behind obstacles (Mov. S2, S3). By compensating for the

respective shortcomings of Lidar and video tracking and using each to complement the trajectories of the other, a tracking system for small, high-speed objects in three dimensions was realized.

Revised manuscript: Page 15, lines 264-267,

A similar 3D tracking system using Kinect sensor or cameras has been previously reported for various animal species, including marmosets^{1, 20, 23, 24, 30, 31}. However, none of the existing 3D tracking systems have achieved such a long-term analysis, whereas FulMAI is an only behavioral analysis system for longitudinal observation over each marmoset's lifetime in the family.

Revised manuscript: Page 16, lines 272-277,

Our system has successfully tracked multiple marker-less marmosets for the first time, which is another advantage of our system over existing systems. The color marking causes animal stress because it requires capture or anesthesia, and periodic hair dyeing requires time and effort, which is unsuitable for analyzing marmosets' behaviors in the long-term. The marmosets face identification, which we were able to implement for the first time, can also work alone and can be applied to other motion analysis systems, such as DeepBhvTracking²⁷.

Revised manuscript: Page 19, lines 327-332,

In addition, some marmoset facilities, to provide more space or different environments for marmosets, occasionally connect adjacent marmosets' cages³⁴. In this study, an extra cage was connected to the home cage. When a marmoset went off-screen due to movement to the connected cage, the tracking of the marmoset was temporarily interrupted. However, other marmosets continued to be tracked, and no other individual was tracked instead of the missing marmoset (Mov. S7). Tracking was then resumed when the marmoset returned to the cage.

Revised manuscript: Page 20, lines 354-356,

For more flexible behavior extraction, this system was made of Robot operating system (ROS) and is extensible to incorporate other Python-based projects, such as DeepLabCut, DeepEthogram³⁶, and TweetyNet³⁷.

Revised manuscript: Page 21, lines 361-364,

Moreover, Lidar can track the locations of animals' day and night; therefore, their nocturnal behavior can be observed to obtain primary data. Our preliminary data showed that the combination with a Lidar and starlight camera can capture details of nighttime behavior in a dark room (Supplemental Mov. S8).

4. I am a bit confused about the process of assessing the accuracy of the system using grooming. I agree with the point that grooming is optimal behavior for assessing accuracy because the proposed system has a higher spatial resolution. The system seemed to detect the distance between individuals and decide if the two were close enough. Human observers probably judge the frame as the grooming by multiple factors, not only the distance. Then, the distance between marmosets is not a sufficient condition for grooming, and there could be a moment the distance is minimal but not deemed as grooming by the observer. Indeed, the authors briefly described this (line222-). However, I think this is not the accuracy of the system but the description of marmoset behavior (i.e., how frequently they groom each other when they stay closer?). The real test of the system is if the grooming and non-grooming behavior by the subtle difference of distance between animals. For example, if grooming or other (e.g., just touching the shoulder) can be dissociated by the system, then I think the system is accurate enough to detect the grooming.

Thank you for your very important comment. We added Supplemental Fig. 6, which shows our system overview, including how the system detects the behavior. We have also added detailed descriptions of the detection of grooming behavior in the material methods section (Page27, lines 481-485). For the grooming detection, Yolo learned the grooming behavior, irrespective of the distance between the

marmosets. As you have mentioned, marmosets are often close together without grooming each other. As we explained to Reviewer 1, for preparing training images to detect grooming behavior, we strictly categorized grooming images and two marmosets close to each other by skilled animal technicians to exclude the incidences from our dataset when marmosets did not groom. Our study confirms that close conditions were not always detected as grooming. In the revised Fig. 6b, we tested for detection of the situation in which marmosets were close to each other but not grooming. The yellow highlight columns in the revised Fig. 6b indicate the incidences when marmosets were close to each other but were not grooming (Fig. 6b).

Revised manuscript: Page 13, lines 220-222,

In one hour, there were 10 units when the distances between marmosets were close, and false positives were detected in only one unit, with a duration of 30 sec (recall: 90%, Fig. 6b).

Revised manuscript: Page 24, lines 429-438,

System workflow

The overview of this system is shown in Supplemental Fig. 6. The videos and Lidar data were processed on the same PC, and acquisition times were perfectly synchronized for at least one month. Yolo, an object recognition algorithm, detected the marmoset's face, body, actions, and enrichments (ball and hammock) in each video frame. Body coordinates were combined with coordinates by Yolo and Lidar to calculate a short 3D trajectory. Furthermore, the cropped face images were identified as individuals by a convolutional neural network, and individual IDs were applied to the short-term 3D trajectories to achieve long-term 3D tracking of each individual. Behaviors were detected simultaneously with the same model of Yolo, and each animal's behavior was linked to the nearest marmoset ID.

Original manuscript:

As training images, 252 annotated image files in which grooming marmosets were surrounded by a minimum rectangle were prepared, and the model was trained using the dataset.

Revised manuscript: Page 27, lines 481-485,

A total of 252 annotated image files in which grooming marmosets were surrounded by a minimum rectangle were prepared as training images, and the model was trained using the dataset. Grooming images were strictly categorized by skilled animal technicians to exclude situations in which marmosets did not groom.

Original manuscript: Fig. 6 Figure legend:

(a) Each cell represented 30 s. Blue cells: Visual observation detected grooming at least once in 30 s. Red cells: The system detected grooming by the intensity exceeding a threshold value.

Revised manuscript: Fig. 6 Figure legend: Page 32, lines 566-568,

(b) Each cell represented 30 s. Blue cells: Visual observation detected grooming at least once in 30 s. Yellow cells: Visual observation detected the close distance between two marmosets. Red cells: The system detected grooming by the intensity exceeding a threshold value.

5. Authors seemed to develop this technology exclusively for the common marmoset. Still, I think this paper will attract more general readers if it could be applied to other species. How advantageous of this system (compared to other systems) in its application to other animals?

Thank you for this great suggestion. We think that this system also can be adopted to other species. Although we developed this system specific to detect marmoset behavior, the system can be modified to train data for other species, because this system uses supervised learning, and Lidar will work regardless of animal species. Accordingly, we have added the following sentence to the discussion.

Revised manuscript: Page 15-16, lines 267-271,

One of the FulMAI's advantage is the flexibility in terms of cage size and structure as long as acrylic

panels are located in front of the camera and Lidar, and it can be applied to a wide variety of animal species. This system would also be useful for other animal species that move as quickly as marmosets in 3D, such as tamarins, free-swimming fish, and quick-flying birds and bats.

6. Related to this point, a large part of the introduction and discussion is about the advantage of the markerless motion capture system in general, not specifically for the developed system per se, for the marmoset. Please limit discussion specifically about (dis)advantage of the developed system over the other system(e.g. see comment 3).

Thank you for your comment. We did not intend to criticize other related methods, but to describe the characteristics of each method to compare the existing systems with our system. In the discussion, we have also revised the text mentioning other studies and our study.

Original manuscript: Deleat

In mice, there are some systems for analyzing animal behavior during free-ranging activities under group-housed conditions. However, unlike mice, marmosets move quickly in three dimensions (3D), thus mouse-specific systems cannot be directly applied to marmosets.

Revised manuscript: Page 4, lines 60-73,

The various systems used for analyzing animal behaviors under free-moving activities include systems suitable for three dimensions (3D) tracking and behavior classification (DANNCE¹⁸, FreiPose¹⁹, and MarmoDetecotor²⁰), for generating pose estimation (DeepLabCut, SLEAP²¹, MARS²², DANNCE, FreiPose, and OpenMonkeyStudio¹), for analyzing behavior in home cages (DeepLabCut^{23, 24}, B-SoiD¹⁷, SLEAP, and MARS), for analyzing multiple animals (DeepLabCut, MiceProfier²⁵, SLEAP, and OpenMonkeyStudio), for finding novel behavioral abnormalities by unsupervised clustering (MoSeq²⁶, B-SoiD, and FreiPose¹⁹), and for use across animal species (DeepLabCut, DeepBhvTracking²⁷, B-SoiD, and DANNCE). These systems have been generally developed for mice; only DeepLabCut, DeepBhvTracking, DANNCE, and MarmoDetector²⁰ have been adapted to marmosets, which move relatively quickly in 3D. A previous study used the pose estimation system to analyze the behavior of two marmosets for nine hours using individual identification markers²³. These analysis systems have observed only a small proportion of an animal's life and longitudinal observation over each marmoset's lifetime (15 years) or more than months of observing a family have not been achieved.

Revised manuscript: Page 5, lines 78-91,

In this study, we developed a behavioral analysis system named FulMAI (Full Monitoring and Animal Identification), that can record and analyze behavior of an animal living among multiple animals over their lifetime. The concept of this system is to detect changes in behavior over an individual's lifespan and understand the relation of changes in brain function. FulMAI is a 3D tracking system that uses cameras, light detection and ranging (Lidar) devices, and deep learning to simultaneously track a marmoset family of three individuals in one cage. Furthermore, to take advantage of stress-free behavior analysis in the home cage, we also developed a system for individual identification via facial recognition using deep learning. The FulMAI system can analyze the location and the period each animal stayed in the cage and the distance between individuals. Furthermore, the time spent in social behavior is an index for analyzing animal interactions. Therefore, as an example of social behavior detection, we developed automatic grooming behavior detection system using deep learning. The technology developed in this study would be applicable in detecting important behavioral changes caused by development, aging, and various diseases in small, nonhuman model primates.

Revised manuscript: Page 16, lines 274-279,

Our system successfully tracked multiple marker-less marmosets for the first time, which is another advantage of our system over existing systems. The color marking causes animal stress because it requires capture or anesthesia, and periodic hair dyeing requires time and effort, which is unsuitable for analyzing marmosets' behaviors in the long-term. The face identification of marmosets, which we

were able to achieve for the first time, can also be undertaken independently and can be applied to other motion analysis systems, such as DeepBhvTracking²⁷.

Revised manuscript: Page 19, lines 331-336,

In addition, some marmoset facilities, to provide more space or different environments for marmosets, occasionally connect adjacent marmosets' cages³⁴. In this study, an extra cage was connected to the home cage. When a marmoset went off-screen due to movement to the connected cage, the tracking of the marmoset was temporarily interrupted. However, other marmosets continued to be tracked, and no other individual was tracked instead of the missing marmoset (Mov. S7). Tracking was resumed when the marmoset returned to the cage.

Revised manuscript: Page 20, lines 350-364,

This finding was obtained exclusively using automatic behavior analysis. While there are existing methods for automatically detecting the behavior of a single animal³⁵, for the first time, our system was able to directly detect the behavior of multiple animals in captivity and identify the individuals and their locations. We aim to identify a variety of behaviors for analyzing lifelong behavior changes. To find unpredictable unknown behavior, clustering by unsupervised learning would be necessary. For unsupervised learning, it is necessary to analyze the videos of each individual or to create a skeletal model of each marmoset and analyze the movement of each key point. Our study is expected to further develop in the future because of its ability to detect accurate 3D location linked to each individual. For more flexible behavior extraction, this system was made of Robot operating system (ROS) and is extensible to incorporate other Python-based projects, such as DeepLabCut, DeepEthogram³⁶, and TweetyNet³⁷. Because all animal activities are recorded in the FulMAI, each animal behavior in this home cage can be analyzed retrospectively in detail. By comparing with other data, such as imaging, blood biochemistry, or biomarker data, it is also possible to analyze the behavior of model animals from multiple perspectives without overlooking changes in behavior associated with the life events of brain function changes or the onset of disease.

7. Similarly, the author's argument may be biased toward the potential analysis of the disease model. I think the combination of face recognition and Lidar could be applicable for general behavioral analysis (as demonstrated in this work), not specifically disease models. Plus, the supervised learning algorithm using this paper cannot detect the unpredictable behavioral phenotype of the marmoset disease model. Therefore, to make this paper more focused, I would recommend being more modest about this (e.g., Lines 60-68; 355-372.)

Thank you for your suggestion. Since the ultimate purpose of this system was to analyze disease model by longitudinal observation of disease model marmosets to understand how to develop and progress the disease, our description may have been a little biased toward disease model analysis. Using marmosets as model animals has advantages over rodents and may help elucidate basic gene functions in the brain, clarify molecular mechanisms underlying neurological disease onset using genetic modifications, and aid in implementing cognitive neuroscience techniques in primate research. As you mentioned, this system is also useful for general behavioral analysis, that is very important for our system to understand normal marmoset behavioral changes during development and aging. Since the introduction was too descriptive of the disease model, we have reduced and modified it to align with developing a behavioral analysis system. As per your comment, our system cannot detect unpredictable abnormal behavior so far. We have added the information that unsupervised learning should be incorporated for detection of unpredictable behaviors.

Original manuscript:

Neurological disorders, such as neurodegenerative diseases and psychiatric disorders, are associated with decline in cognitive function and increases incidence of abnormal behaviors, which fundamentally affect social life. Mouse models, including genetically modified mice, have contributed to our understanding of brain structures, elucidating mechanisms of the onset and progression of neuronal disease, and behavioral changes caused by diseases. However, because of the anatomical and

physiological differences between rodents and humans, mouse models do not fully recapitulate the pathogenesis of neuronal diseases. Therefore, there are limitations to using mouse models to address questions around human brain function. In contrast, nonhuman primates do share anatomical and physiological features with humans, especially in specialized functions acquired through unique cerebral expansions, such as tool use, language, and self-awareness. As these functions are particularly affected in neurodegenerative diseases and psychiatric disorders, nonhuman primate models are crucial for understanding the development of neuronal and psychiatric disorders.

The common marmoset (*Callithrix jacchus*) is a small nonhuman primate with behavioral and social characteristics resembling those of humans. They are diurnal, form monogamous family units, engage in altruistic behaviors such as food sharing, and cooperate with all family members to raise their youngest offspring. As marmoset body size and social units are small, maintaining complete social units for research in a laboratory is easier than that for other primates. Therefore, marmosets are an ideal model for studying social behaviors in terms of maintaining families as social units. Recent advances in genetic modification techniques in nonhuman primates have enabled the production of several genetically modified disease-model marmosets, such as those for neurodegenerative diseases. Using a marmoset model has advantages over rodents as model animals and may help elucidate basic gene functions in the brain, clarify molecular mechanisms underlying neurological disease onset using genetic modifications, and aid in implementing cognitive neuroscience techniques in primate research. For these purposes, national brain research projects using marmosets have been conducted in the US and Japan.

Revised manuscript: Page 3, lines 33-50,

Behavioral analyses are important for understanding brain function changes during development and aging and for evaluating disease onset in model animals. In particular, nonhuman primates are often used to study brain function, and behavioral characteristics provide an indispensable source of data for hypothesis testing¹. Nonhuman primates share anatomical and physiological features with humans, especially in specialized functions acquired through unique cerebral expansions, such as tool use, language (for example, vocal communication), and self-awareness^{2, 3, 4, 5}. The common marmoset (*Callithrix jacchus*) is a small nonhuman primate with behavioral and social characteristics resembling those of humans. They are diurnal, form monogamous families, engage in altruistic behaviors such as food sharing, and cooperate with all family members to raise their youngest offspring^{3, 6, 7}. As marmoset body size and social units are small, maintaining complete social units for research in a laboratory is easier than that for other primates. Therefore, marmosets are an ideal model for studying social behaviors in social units³. Recent developments in genetic modification techniques in marmosets have enabled the production of various disease-model marmosets, such as those for neurodegenerative diseases^{8, 9, 10}. Using marmosets as model animals has advantages over rodents and may help elucidate basic gene functions in the brain, clarify molecular mechanisms underlying neurological disease onset using genetic modifications, and aid in implementing cognitive neuroscience techniques in primate research^{5, 11}. For these purposes, national brain research projects using marmosets have been conducted in the US and Japan^{12, 13}.

Revised manuscript: Page 20, lines 354-360,

To find unpredictable unknown behavior, clustering by unsupervised learning would be necessary. For unsupervised learning, it is necessary to analyze the videos of each individual or to create a skeletal model of each marmoset and analyze the movement of each key point. Our study is expected to further develop in the future because of its ability to detect accurate 3D location linked to each individual. For more flexible behavior extraction, this system was made of Robot operating system (ROS) and is extensible to incorporate other Python-based projects, such as DeepLabCut, DeepEthogram³⁶, and TweetyNet³⁷.

Minor comments,

1. Please provide detailed information as to the installation of Lidar (e.g., the distance between, the height from the floor) and the required specification of the PC system, and the extent of error of measurement.

Thank you for your suggestion. We have added detailed information in the materials and methods section.

Revised manuscript: Page 22, lines 394-395,

The lower Lidar was set at 0.9 m height and the upper Lidar at 1.9 m height. The distance between the left and right Lidar devices was 0.3 m.

Revised manuscript: Page 23, lines 402-403,

The specifications of the PC used to operate and record the camera and Lidar were corei7, RTX1060, 32GB. PC specifications used for analysis were corei7, RTX3090, 64 GB.

2. If authors will refer to the usage of this system in the dark (nighttime), please describe if the proposed system could be applicable for that purpose.

Lidar can continue to track marmosets at night as it does during the day. Marmosets normally sleep at all night, but, for example, the Parkinson's marmoset model is known to cause sleep disturbances. We are planning to update the tracking system to combine Lidar with a starlight camera, and to detect such sleep disturbances and record behavioral details. Supplemental Mov. S8 shows a nighttime tracking video taken with our developing device.

Revised manuscript: Page 21, lines 365-368,

Moreover, Lidar can track the animal locations day and night; therefore, their nocturnal behavior can be observed to obtain primary data. Our preliminary data showed that the combination with a Lidar and starlight camera can capture details of nighttime behavior in a dark room (Supplemental Mov. S8).

3. Line 35-59. Introduction. The first two sentences can be eliminated because this is a technical paper about motion capturing.

As you suggested, the introduction of gene-modified mouse and neurological disorders diverged from the main text; therefore, we removed this sentence.

Revised manuscript: Page 3, lines 36-50,

Nonhuman primates share anatomical and physiological features with humans, especially in specialized functions acquired through unique cerebral expansions, such as tool use, language (for example, vocal communication), and self-awareness^{2, 3, 4, 5}. The common marmoset (*Callithrix jacchus*) is a small nonhuman primate with behavioral and social characteristics resembling those of humans. They are diurnal, form monogamous families, engage in altruistic behaviors such as food sharing, and cooperate with all family members to raise their youngest offspring^{3, 6, 7}. As marmoset body size and social units are small, maintaining complete social units for research in a laboratory is easier than that for other primates. Therefore, marmosets are an ideal model for studying social behaviors in social units³. Recent developments in genetic modification techniques in marmosets have enabled the production of various disease-model marmosets, such as those for neurodegenerative diseases^{8, 9, 10}. Using marmosets as model animals has advantages over rodents and may help elucidate basic gene functions in the brain, clarify molecular mechanisms underlying neurological disease onset using genetic modifications, and aid in implementing cognitive neuroscience techniques in primate research^{5, 11}. For these purposes, national brain research projects using marmosets have been conducted in the US and Japan^{12, 13}.

4. Supplementary video is too large in size. It took minutes to download. It would be helpful to edit and extract the specific timeline.

We apologize for the large video size. We wanted you to see as much information as possible; therefore, we made this video longer. We will resubmit the video in a smaller size. We split the video file into multiple files.

REVIEWERS' COMMENTS:

Reviewer #1 (Remarks to the Author):

Terumi Yurimoto et al. addressed the majority of my suggestions of my first review. The few points that remain can be addressed in a minor revision.

1. Please put Figure S7 into the main figures, it will help the readers to understand the experiment procedure.
2. The authors claimed that the purpose of the FulMAI was to track the individual animal's behaviors for a long time (even for tens of years). However, there is no any data to show the result of long time comparison. From my experience, the detectors in the early training stages by YOLO may not work in the later stages. So please add more data or downgrade the claims.
3. What's the wavelength and power of the Lidar, does it harm to animals for the long term usage?

Reviewer #2 (Remarks to the Author):

This is a manuscript that I previously reviewed. The authors have done an excellent job of addressing my previous concerns with the manuscript. However, there are a few additional concerns I would like the authors to address.

Major comments

Lines 33-54: The section discussing marmoset behavior, while informative, is overly redundant. I recommend condensing this paragraph. A brief statement summarizing the marmoset's advantages, such as its short lifecycle and social characteristics, would suffice. Additionally, it's unclear why the emphasis is placed predominantly on "brain function" and "disease" onset in this context. Behavioral analysis encompasses a broad spectrum of biological, psychological, and ethological studies in both animals and humans. If the focus is to highlight the value of this system in the context of brain function or disease progression, it would be beneficial to include data demonstrating its effectiveness in assessing these aspects.

2. Lines 237-240, 368-371,384. See above.

Minor comments

Line 34: The term "disease onset" could be replaced with "progression of disease" for clarity.

Line 118 and subsequent mentions: The naming convention for the animals (I5072M/father, I5894F/mother, I940M/juvenile) is somewhat confusing. A simpler approach, such as using labels like A, B, and C for the father, mother, and juvenile respectively, might be easier to follow, especially with just three animals.

Line 224: The addition of grooming behavior analysis is a positive aspect, but the corresponding video in Mov. S6 does not clearly depict the behavior due to its small size. Providing a larger or more distinct image would enhance the understanding of this behavior.

REVIEWERS' COMMENTS:

Reviewer #1 (Remarks to the Author):

Terumi Yurimoto et al. addressed the majority of my suggestions of my first review. The few points that remain can be addressed in a minor revision.

Thank you for your reviewing again. Our correction to your points is below.

1. Please put Figure S7 into the main figures, it will help the readers to understand the experiment procedure.

Thank you for your suggestion. We agree with your suggestion and have moved Figure S7 to the main Fig. 1c and the description of the system workflow from Materials and Methods to Results. The following is a list of modifications.

Revised manuscript: Page 7, lines 109-118

The overview of this system workflow is shown in Fig. 1c. The videos and Lidar data were processed on the same PC, and acquisition times were perfectly synchronized for at least one month. Yolo, an object recognition algorithm, detected the marmoset's face, body, actions, and environmental enrichments (ball and hammock) in each video frame. Body coordinates were combined with coordinates using Yolo and Lidar to calculate a short 3D trajectory. Furthermore, the cropped face images were identified as individuals using a convolutional neural network, and individual IDs were applied to the short-term 3D trajectories to achieve long-term (average 2.8 min.) 3D tracking of each individual. Behaviors were detected simultaneously with the same Yolo model, and each animal's behavior was linked to the nearest marmoset ID.

Original manuscript:

Figure 1. Installation of hardware for the behavior analysis system

(a) A home cage with acrylic panels on the front and back and metal mesh on the sides. (b) Schematic diagram of light and ranging (Lidar) systems and cameras installed in front of the cage. Lidars and cameras were installed 1000 mm in front of the cage.

Revised manuscript: Page 31, lines 534-541

Figure 1. Installation of hardware and software for the FulMAI system

(a) A home cage with acrylic panels on the front and back and metal mesh on the sides. (b) Schematic diagram of light and ranging (Lidar) systems and cameras installed in front of the cage. Lidars and cameras were installed 1000 mm in front of the cage. (c) The four videos were assigned boxes and labels by Yolo. The face label parts were used for face identification, and the body portion was used for 3D tracking with Lidar centroid. The same detector was also used to detect grooming behavior. Finally, face identification and 3D tracking were linked to obtain the 3D trajectory of each marmoset and the 3D coordinates of the behavior.

2. The authors claimed that the purpose of the FulMAI was to track the individual animal's behaviors for a long time (even for tens of years). However, there is no any data to show the result of long time comparison. From my experience, the detectors in the early training stages by YOLO may not work in the later stages. So please add more data or downgrade the claims.

Thank you very much for your comment. Regarding the marmoset body detection, the Mov. 1, 4, and 5 shown in the original text were acquired at 5-month intervals, while Mov. 1 and 5 have a 10-month interval. In these movies, we used the same Yolo-trained model during this period without any problem. Therefore, we believe there is no problem with Yolo's behavior for longitudinal analysis and for long periods. We added Supplementary Fig. 6, in which we acquired data for several months from another individual. On the other hand, for face identification, periodic relearning was necessary (once a month during this experimental period). Since these data were available for a long period, we think they can be applied to longer periods than previously reported. We have added a description of the maximum number of days in the Results, a possible reason why FulMAI can be used for a longer period in the Discussion, and model updates for face identification in the Materials and Methods.

However, as you pointed out, we have not run the system for several decades. Therefore, we have changed “long-term” to “longitudinal.”

Original manuscript:

The common marmoset (*Callithrix jacchus*), that lives in a nuclear family like humans, is a useful model, but long-term automated behavioral observation of multiple animals has not been achieved. Here, we developed a Full Monitoring and Animal Identification (FulMAI) system for long-term detection of three-dimensional (3D) trajectories of each individual in multiple marmosets under free-moving conditions by combining video tracking, Light Detection And Ranging, and deep learning.

Revised manuscript: Page 2, lines 22-26

The common marmoset (*Callithrix jacchus*), which lives in a nuclear family like humans, is a useful model, but longitudinal automated behavioral observation of multiple animals has not been achieved. Here, we developed a Full Monitoring and Animal Identification (FulMAI) system for longitudinal detection of three-dimensional (3D) trajectories of each individual in multiple marmosets under free-moving conditions by combining video tracking, Light Detection and Ranging, and deep learning.

Original manuscript:

The same analyses were performed for long-term analysis on a different day during the daytime, 7:00 am–7:00 pm. After 6:00 pm, all marmosets moved to the bed and clustered together and their faces were not observed; therefore, the data were excluded from the analysis.

Revised manuscript: Page 10, lines 159-161

The same analyses were performed on a different day during the daytime, 7:00 am–7:00 pm. After 6:00 pm, all marmosets moved to the bed, and clustered together and their faces were not observed; therefore, the data were excluded from the analysis.

Revised manuscript: Page 10, lines 178-181

Although a similar analysis was performed for 76 days, the maximum continuous analysis period for this family was 35 days because of a downtime period due to system maintenance (Supplementary Fig. 6a). In another marmoset family, this system ran for four months continuously and was able to obtain longitudinal data (Supplementary Fig. 6b).

Original manuscript:

Based on these findings, FulMAI is expected to record the lifetime behavior of marmosets with a lifespan of approximately 15 years and match it with MRI and other data to determine what changes occurred in the brain during life events and transformed their behavior.

Revised manuscript: Page 14, lines 244-246

FulMAI is expected to record and analyze longitudinal marmoset behavior and reveal what behavioral changes occur within the same individual before and after life events.

Original manuscript:

In addition, because both Lidar and video tracking can be processed at high speed without a time lag, data is not accumulated for subsequent analysis, which is favorable for long-term analysis. Indeed, the 3D trajectories and grooming behaviors of three marmosets could be continuously analyzed in their home cage for over six months without data accumulation.

Revised manuscript: Page 15, lines 267-271

In addition, because both Lidar and video tracking can be processed at high speed without a time lag, data is not accumulated for subsequent analysis, which is favorable for longitudinal analysis. Indeed, the 3D trajectories and grooming behaviors of three marmosets could be continuously analyzed in their home cage for over one month without data accumulation (Supplementary Fig. 6a).

Original manuscript:

A similar 3D tracking system using Kinect sensor or cameras has been reported for various animal species, including marmosets^{1, 20, 23, 24, 30, 31}. However, none of the existing 3D tracking systems have

achieved such a long-term analysis, whereas FulMAI is the only behavioral analysis system for longitudinal observation over each marmoset's lifetime in the family.

Revised manuscript: Page 16, lines 273-281

Although a similar 3D tracking system using Kinect or cameras has been reported for various animal species, including marmosets^{1,11,14,15,21,22}, none of the existing multiple 3D tracking systems have achieved such a continuous analysis for a month. Mov.1, 4, and 5 show the same marmoset analysis data from the continuous FulMAI data at 5-month intervals. Yolo was trained only the first time during this period, and continuously obtained 3D tracking data without any problems. In addition, tracking information could be obtained for several consecutive months (Supplementary Fig. 6). Therefore, Yolo and Lidar are thought that no maintenance is required. On the other hand, face identification required monthly training. FulMAI would be able to analyze longitudinal behavioral changes over each marmoset's lifetime in the family.

Original manuscript:

Incorporating facial recognition results into the trajectory enabled long-term tracking, even for multiple animals in the same cage (Fig. 3).

Revised manuscript: Page 18, lines 311-312

Incorporating facial recognition results into the trajectory enabled longitudinal tracking, even for multiple animals in the same cage (Fig. 3).

Revised manuscript: Page 26, lines 460-461

For longitudinal analyses, training data collection and training were performed once a month to update the model.

3. What's the wavelength and power of the Lidar, does it harm to animals for the long term usage?

Thank you very much a very important comment. It is very important if the devices affect the health and behavior of the marmosets. The Lidar, VLP-16, uses a laser classified as a Class 1 approved by the FDA, and its wavelength is 903 nm. It has been two years since we installed this laser in front of the cage, and no apparent dysfunction has appeared in the marmosets. We added VLP-16 wavelengths and classes of lasers to Materials and Methods, and health effects to Discussion.

Original manuscript:

The animals were healthy throughout the experiment, proving the safety of our system.

Revised manuscript: Page 15, lines 271-272

The animals were healthy throughout the experiment and had no apparent dysfunction caused by Lidar, proving the safety of our system.

Revised manuscript: Page 23, lines 404-406

The Lidar utilizes a 903nm infrared laser light, which is eye-safe (Class I laser) and a reported range detection of up to 100 meters.

Reviewer #2 (Remarks to the Author):

This is a manuscript that I previously reviewed. The authors have done an excellent job of addressing my previous concerns with the manuscript. However, there are a few additional concerns I would like the authors to address.

Thanks again for the review. We have described our views on the points you raised.

Major comments

Lines 33-54: The section discussing marmoset behavior, while informative, is overly redundant. I recommend condensing this paragraph. A brief statement summarizing the marmoset's advantages, such as its short lifecycle and social characteristics, would suffice. Additionally, it's

unclear why the emphasis is placed predominantly on "brain function" and "disease" onset in this context. Behavioral analysis encompasses a broad spectrum of biological, psychological, and ethological studies in both animals and humans. If the focus is to highlight the value of this system in the context of brain function or disease progression, it would be beneficial to include data demonstrating its effectiveness in assessing these aspects.

Thank you for your important comment. As you point out, behavioral analysis is used for various purposes, and we expect that FulMAI system will be used for various applications. In marmosets, non-human primates, we believe that there is also a demand for brain function research. While it is not possible to remove all references to brain function and disease, we have removed the sections that focus too much on brain function and moved the section on genetic modification and disease to the Discussion section.

Original manuscript:

Behavioral analyses are important for understanding brain function changes during development and aging and for evaluating disease onset in model animals. In particular, nonhuman primates are often used to study brain function, and behavioral characteristics provide an indispensable source of data for hypothesis testing¹. Nonhuman primates share anatomical and physiological features with humans, especially in specialized functions acquired through unique cerebral expansions, such as tool use, language (for example, vocal communication), and self-awareness^{2, 3, 4, 5}. The common marmoset (*Callithrix jacchus*) is a small nonhuman primate with behavioral and social characteristics resembling those of humans. They are diurnal, form monogamous families, engage in altruistic behaviors such as food sharing, and cooperate with all family members to raise their youngest offspring^{3, 6, 7}. As marmoset body size and social units are small, maintaining complete social units for research in a laboratory is easier than that for other primates. Therefore, marmosets are an ideal model for studying social behaviors in social units³. Recent developments in genetic modification techniques in marmosets have enabled the production of various disease-model marmosets, such as those for neurodegenerative diseases^{8, 9, 10}. Using marmosets as model animals has advantages over rodents and may help elucidate basic gene functions in the brain, clarify molecular mechanisms underlying neurological disease onset using genetic modifications, and aid in implementing cognitive neuroscience techniques in primate research^{5, 11}. For these purposes, national brain research projects using marmosets have been conducted in the US and Japan^{12, 13}.

Revised manuscript: Page 3, lines 33-43

Behavioral analyses are important in broad research areas such as animal and human biological, psychological, and ethological studies. In neuroscience research, behavioral analysis is one of the most important analytical methods for understanding changes in brain function during development and aging and assessing disease development in animal models. In particular, nonhuman primates are often used to study brain function, and behavioral characteristics provide an indispensable source of data for hypothesis testing¹. The common marmoset (*Callithrix jacchus*) is a small nonhuman primate with behavioral and social characteristics resembling those of humans. They are diurnal, form monogamous families, engage in altruistic behaviors such as food sharing, and cooperate with all family members to raise their youngest offspring²⁻⁴. As marmoset body size and social units are small, maintaining similar to social units in the wild for research laboratory is easier than that for other primates. Therefore, marmosets are an ideal model for studying social behaviors in social units².

Original manuscript:

To evaluate long-term brain function changes due to development, aging, and progression of disease using marmosets as a model, it is important to capture changes in behavior over a lifespan, and an analysis of multiple marmosets under free-moving conditions is necessary to capture changes in social behavior.

Revised manuscript: Page 3, lines 44-47

To evaluate long-term behavioral changes due to physiological changes, such as development, aging, and progression of disease using marmosets, it is important to capture changes in behavior over a lifespan, and an analysis of multiple marmosets under free-moving conditions is necessary to capture changes in social behavior.

2. Lines 237-240, 368-371,384. See above.

Thank you for your comment. Similar to your above point, we have revised the limitation to brain function and disease so that it is not limited. On the other hand, we believe that it can also be used for application to disease models, so we have left part of it.

Original manuscript:

FulMAI is expected to record the lifetime behavior of marmosets with a lifespan of approximately 15 years and match it with MRI and other data to determine what changes occurred in the brain during life events and transformed their behavior.

Revised manuscript: Page 14 , lines 244-246

FulMAI is expected to record and analyze longitudinal marmoset behavior and reveal what behavioral changes occur within the same individual before and after life events.

Original manuscript:

Sleep disorders have recently been suggested as predictors of Alzheimer's disease and Parkinson's disease^{38,39}. This system would be able to accurately quantify the behavior of marmosets throughout their lives to discover behavioral changes before and after the onset of diseases, even at an extremely early stage.

Revised manuscript: Page 21 , lines 381-383

Sleep disorders have recently been reported as predictors of Alzheimer's disease and Parkinson's disease^{30,31}. Recently, we have established two kinds of Alzheimer's disease models²⁵ and expect to observe sleep disorders with this Lidar and starlight camera system.

Original manuscript:

These measurements will allow automated analysis of activity and behavior under free-moving conditions throughout the marmoset's lifetime, enabling quantitative analysis of "when, where, and what" of each animal and capturing changes in brain function, including development, growth, aging, and disease.

Revised manuscript: Page 22 , lines 395-398

These measurements will allow automated analysis of activity and behavior under free-moving conditions throughout the marmoset's lifetime, enabling quantitative analysis of "when, where, and what" of each animal and capturing changes of development, growth, aging, and disease.

Minor comments

Line 34: The term "disease onset" could be replaced with "progression of disease" for clarity.

Thank you for pointing this out. we have replaced the term as you indicated.

Revised manuscript: Page 3 , lines 44-47

To evaluate long-term behavioral changes due to physiological changes, such as development, aging, and progression of disease using marmosets, it is important to capture changes in behavior over a lifespan, and an analysis of multiple marmosets under free-moving conditions is necessary to capture changes in social behavior.

Line 118 and subsequent mentions: The naming convention for the animals (I5072M/father, I5894F/mother, I940M/juvenile) is somewhat confusing. A simpler approach, such as using labels like A, B, and C for the father, mother, and juvenile respectively, might be easier to follow, especially with just three animals.

Thank you for your comment. We have replaced the name of the marmoset with the following: I5072M/father is marmoset A, I5894F/mother is marmoset B, and I940M/juvenile is marmoset C. The revised area is highlighted in green to distinguish it from other revisions.

Original manuscript:

All 3D tracking and grooming behavior detection tests were conducted using data from one family

consisting of three marmosets (I5072M/father (I5072M/f), I5894F/mother (I5894F/m), and I940M/juvenile (I940M/j)) in a cage.

Revised manuscript: Page 7, lines 118-120

All 3D tracking and grooming behavior detection tests were conducted using data from one family consisting of three marmosets (I5072M/father (marmoset A), I5894F/mother (marmoset B), and I940M/juvenile (marmoset C)) in a cage.

Original manuscript:

This data indicated that I5072M/f stayed on the lower floor of the cage for 88.8% (23642/26626 frames) of the one hour, whereas both I5894F/m and I940M/j stayed on the lower floor for less than 0.5% (128/25597 and 12/24234 frames) of the one hour. In contrast, they stayed on middle floor for half of the one hour and actively moved in various areas in the cage except the lower floor rest of the time (Fig. 4a-c, Table 3, Supplementary Fig. 1).

Revised manuscript: Page 9, lines 146-150

This data indicated that marmoset A stayed on the lower floor of the cage for 88.8% (23642/26626 frames) of the one hour, whereas both marmoset B and marmoset C stayed on the lower floor for less than 0.5% (128/25597 and 12/24234 frames) of the one hour. In contrast, they stayed on middle floor for half of the one hour and actively moved in various areas in the cage except the lower floor rest of the time (Fig. 4a-c, Table 3, Supplementary Fig. 1).

Original manuscript:

During this time, the distance between I940M/j and I5894F/m was less than 0.5 m 25.9% (5044/19501 frames) of the total time. However, a distance of less than 0.5 m between I940M/j-I5072M/f and I5072M/f-I5894F/m was detected 1.0% (203/20409 frames) and 2.2% (475/21599 frames) of the time, respectively. Conversely, the distance between I940M/j-I5894F/m was more than 1 m 2.0% (391/19501 frames) of the time, whereas I940M/j-I5072M/f and I5072M/f-I5894F/m were separated by more than 1 m distance 20.0% (4081/20409 frames) and 11.0% (2364/21599 frames) of the time, respectively. In this family, I940M/j and I5894F/m spent more time together than I5072M/f.

Revised manuscript: Page 9, lines 151-158

During this time, the distance between marmoset C and marmoset B was less than 0.5 m 25.9% (5044/19501 frames) of the total time. However, a distance of less than 0.5 m between marmoset A-marmoset C and marmoset A-marmoset B was detected 1.0% (203/20409 frames) and 2.2% (475/21599 frames) of the time, respectively. Conversely, the distance between marmoset B-marmoset C was more than 1 m 2.0% (391/19501 frames) of the time, whereas marmoset A-marmoset C and marmoset A-marmoset B were separated by more than 1 m distance 20.0% (4081/20409 frames) and 11.0% (2364/21599 frames) of the time, respectively. In this family, marmoset B and marmoset C spent more time together than marmoset A.

Original manuscript:

The detailed analyses indicated that, among the family members, I940M/j stayed on the lower floor of the cage the longest, unlike the results of the 1-hour analysis (Supplementary Fig. 2). The time spent on the lower floor of the cage between 7:00 am and 6:00 pm for each individual was 8.6% for I5072M/f, 13.0% for I5894F/m, and 23.5% for I940M/j. Throughout the day, I940M/j spent the most time on the middle floor (27.7%), I5072M/f on the upper floor (29.7%), and I5894F/m on the upper floor (29.9%).

Revised manuscript: Page 10, lines 161-166

The detailed analyses indicated that, among the family members, marmoset C stayed on the lower floor of the cage the longest, unlike the results of the 1-hour analysis (Supplementary Fig. 2). The time spent on the lower floor of the cage between 7:00 am and 6:00 pm for each individual was 8.6% for marmoset A, 13.0% for marmoset B, and 23.5% for marmoset C. Throughout the day, marmoset C spent the most time on the middle floor (27.7%), marmoset A on the upper floor (29.7%), and marmoset B on the upper floor (29.9%).

Original manuscript:

I5072M/f also spent most of his time in the upper area in the morning (9:00 am; 54%), but gradually changed his activity area to the middle area in the afternoon (1:00 pm; 51%) and the lower area in the

evening (4:00 pm; 45%). I5894F/m rarely stayed in the middle area but moved to the upper section in the morning (9:00 am; 48%) and to the lower section in the afternoon (3:00 pm; 55%). I940M/j moved most evenly throughout the day among the three in the cage. At night, all individuals moved to the upper bunk to their beds (Supplementary Fig. 2).

Revised manuscript: Page 10, lines 168-173

Marmoset A also spent most of his time in the upper area in the morning (9:00 am; 54%), but gradually changed his activity area to the middle area in the afternoon (1:00 pm; 51%) and the lower area in the evening (4:00 pm; 45%). Marmoset B rarely stayed in the middle area but moved to the upper section in the morning (9:00 am; 48%) and to the lower section in the afternoon (3:00 pm; 55%). Marmoset C moved most evenly throughout the day among the three in the cage. At night, all individuals moved to the upper bunk to their beds (Supplementary Fig. 2).

Original manuscript:

The results showed that I5072M/f-I5894F/m were closer than 0.5 m for 21.1% of the daytime. However, I5072M/f-I940M/j and I5894F/m-I940M/j were closer than 0.5 m for 19.6% and 16.7%, respectively.

Revised manuscript: Page 10, lines 174-177

The results showed that marmoset A-marmoset B were closer than 0.5 m for 21.1% of the daytime. However, marmoset A-marmoset C and marmoset B-marmoset C were closer than 0.5 m for 19.6% and 16.7%, respectively.

Original manuscript:

For example, the male marmoset (I5072M/f) preferred the upper area in the morning, but in the afternoon, it preferred the lower bed.

Revised manuscript: Page 19, lines 327-329

For example, the male marmoset (marmoset A) preferred the upper area in the morning, but in the afternoon, it preferred the lower bed. This result was not apparent in the 1-hour analysis compared with observations.

Original manuscript:

For example, a one-hour analysis showed that I5072M/f preferred the cage's lower floor and remained there 80% of the analysis time (Table 3). However, a 12-hour analysis indicated that I940M/j spent the longest time (23.5%) on the lower floor (Supplemental Fig. 3, 4). The analyses of the distance between individuals showed that I5894F/m-I940M/j were within 0.5 m for 25.9% of the one-hour analysis, but the 12-hour analysis indicated that the I5072M/f-I5894F/m remained closer than I5894F/m-I940M/j or I5072M/f-I5894F/m.

Revised manuscript: Page 19, lines 330-333

For example, a one-hour analysis showed that marmoset A preferred the cage's lower floor and remained there 80% of the analysis time (Table 3). However, a 12-hour analysis indicated that marmoset C spent the longest time (23.5%) on the lower floor (Supplementary Fig. 3, 4). The analyses of the distance between individuals showed that marmoset B-marmoset C were within 0.5 m for 25.9% of the one-hour analysis, but the 12-hour analysis indicated that the marmoset A-marmoset B remained closer than marmoset B-marmoset C or marmoset A-marmoset B.

Original manuscript:

Indeed, when we directly observed this family, I5072M/f and I5894F/m were usually far apart, but a 12-hour analysis suggests that the I5072M/f and I5894F/m spent time close together.

Revised manuscript: Page 19, lines 338-339

Indeed, when we directly observed this family, marmoset A and marmoset B were usually far apart, but a 12-hour analysis suggests that the marmoset A and marmoset B spent time close together.

Original manuscript:

The observed family consisted of a male (I5072M/f, ten years old), female (I5894F/m, six years old), and a juvenile (I940M/j, one year old).

Revised manuscript: Page 19, lines 432-433

The observed family consisted of a male (marmoset A, ten years old), female (marmoset B, six years old), and a juvenile (marmoset C, one year old).

Original manuscript:

The individual I940M/j was born by embryo transfer to the mother's uterus¹⁰, and there was no blood relationship within the third degree of the family.

Revised manuscript: Page 25, lines 437-438

The individual marmoset C was born by embryo transfer to the mother's uterus²⁵, and there was no blood relationship within the third degree of the family.

Original manuscript:

Individual information added to c and d using face identification, green: I5072M/f, red: I5894F/m, yellow: I940M/j.

Revised manuscript: Page 32, lines 547-548

Individual information added to c and d using face identification, green: marmoset A, red: marmoset B, yellow: marmoset C.

Original manuscript:

Green: I5072M/f; red: I5894F/m; yellow: I940M/j.

Revised manuscript: Page 32, lines 552-553

Green: marmoset A; red: marmoset B; yellow: marmoset C.

Original manuscript:

I5072M/f, (b) I5894F/m, (c) I940M/j. (d) Histogram of the distance between each family member. Blue bars indicate the individual distances between I940M/j and I5072M/f, orange bars indicate the individual distances between I940M/j and I5894F/m, and green bars indicate the individual distances between I5072M/f and I5894F/m.

Revised manuscript: Page 32, lines 556-560

marmoset A, (b) marmoset B, (c) marmoset C. (d) Histogram of the distance between each family member. Blue bars indicate the individual distances between marmoset A and marmoset C, orange bars indicate the individual distances between marmoset B and marmoset C, and green bars indicate the individual distances between marmoset A and marmoset B.

Original manuscript:

Red indicates I940M/j, green indicates I5072M/f, and blue indicates I5894F/m.

Revised manuscript: Page 33, lines 564

Green indicates marmoset A, blue indicates marmoset B and red indicates marmoset C

Line 224: The addition of grooming behavior analysis is a positive aspect, but the corresponding video in Mov. S6 does not clearly depict the behavior due to its small size. Providing a larger or more distinct image would enhance the understanding of this behavior.

Thank you for your comment. We apologize for the difficulty in viewing the video. Since there are no animals in the bottom half of the cage in this video, we removed the bottom half and enlarged the video. The video was also edited to leave the upper camera video for better viewing.